# Clathrin-independent endocytosis and retrograde transport in cancer cells tune immune synapse organization and CD8 T cell response

Shiqiang Xu[1,2], Alix Buridant[1], Thibault Hirsch[3], Céline Duhamel[3], Benjamin Ledoux[4], Massiullah Shafaq-Zadah[5], Estelle Dransart[5], Louise Thines[1], Ludger Johannes[5], Pierre Van der Bruggen[3,6], Pierre Morsomme[2], Henri-François Renard[1,4]*

[1]UNamur, Namur Research Institute in Life Sciences, Research Unit in Cell Biology, Namur, Belgium; [2]UCLouvain, Louvain Institute of Biomolecular Science and Technology, Group of Molecular Physiology, Louvain-la-Neuve, Belgium; [3]UCLouvain, de Duve Institute, Bruxelles, Belgium; [4]UNamur, Morph-Im Platform, Namur, Belgium; [5]Institut Curie, PSL Research University, Cellular and Chemical Biology unit, U1143 INSERM, UMR3666 CNRS, Paris, France; [6]WELBIO, Wavre, Belgium

*For correspondence:
henri-francois.renard@unamur.be

## eLife Assessment

This study now provides **solid** evidence for a role of EndoA3-mediated trafficking of ICAM-1 to the immune synapse with T cells. The study will be **valuable** to those studying cell-cell communication in the immune system, and opens additional questions regarding the mechanisms involved and how other adhesion ligands are regulated.

**Abstract** Endophilin A3-mediated clathrin-independent endocytosis (EndoA3-mediated CIE) contributes to the internalization of immunoglobulin-like proteins, including key immune synapse components. Here, we identify ICAM1 as a novel EndoA3-dependent cargo, alongside ALCAM. We demonstrate that both proteins subsequently follow retromer-dependent retrograde transport to the *trans*-Golgi network (TGN) in cancer cells. From there, we propose that they undergo polarized redistribution to the plasma membrane, where they contribute to immune synapse formation between cancer cells and cytotoxic CD8 T cells. Disruption of EndoA3 or retromer components significantly affects the response of autologous cytotoxic CD8 T cells, as evidenced by reduced cytokine production and secretion, but increased lytic activity, while proliferation and later activation marker expression remain intact. This is accompanied by diminished ICAM1 density at the immune synapse, where we observe it arriving via polarized vesicular transport, indicating altered synapse organization. Indeed, cancer cells lacking EndoA3-mediated CIE or retromer form enlarged immune synapses that fail to sustain full T cell cytokine secretion, suggesting a compensatory attempt by T cells to overcome the defective synapse, while likely promoting more transient contacts that potentially favor serial killing. Together, these findings reveal that EndoA3-mediated CIE and retrograde transport act in concert in cancer cells to relocate immune synapse components via the Golgi, thereby fine-tuning the balance between cytotoxic T cell cytokine secretion and lytic activity. These insights contribute to a better understanding of the mechanisms governing immune synapse formation and organization, providing a necessary foundation for the long-term identification of new strategies to enhance T cell–mediated anti-tumor immunity.

## Introduction

Endocytosis is a fundamental physiological process involving membrane-bound carrier-mediated internalization of extracellular substances and cell membrane components into the cytoplasm. This process is essential for maintaining plasma membrane homeostasis and regulating signal transduction. Broadly, endocytosis can be classified into conventional clathrin-mediated endocytosis (CME) and unconventional clathrin-independent endocytosis (CIE). While CME has been extensively characterized, CIE remains less understood and complex as it includes several distinct endocytic mechanisms (*Kaksonen and Roux, 2018*; *Renard and Boucrot, 2021*). Following endocytosis, internalized cargoes are sorted in early/sorting endosomes, where they are either directed toward lysosomes for degradation or recycled to the plasma membrane for further activity. Some are recycled directly to the plasma membrane via Rab4/Rab11 recycling endosomes (*Sönnichsen et al., 2000*; *van der Sluijs et al., 1992*), while others are retrieved by retromer or Commander complexes for transport to the *trans*-Golgi network (TGN), a process termed retrograde transport (*Seaman, 2004*; *McNally et al., 2017*; *Healy et al., 2023*; *Seaman et al., 1998*). Increasing evidence has linked retrograde transport to the maintenance of cell polarity (*Shafaq-Zadah et al., 2020*). For instance, $\beta_1$ integrins are transported to the TGN via the retrograde transport route post-endocytosis, from where they are redistributed in a polarized manner to the leading edge of migrating cells, supporting front-rear polarity crucial for persistent migration (*Shafaq-Zadah et al., 2016*). Similarly, in T cells, adaptor molecules such as the Linker for Activation of T cells (LAT) are transported to the TGN after endocytosis and redistributed to the immune synapse (*Carpier et al., 2018*). The immune synapse is a polarized structure formed between immune and target cells (e.g., cytotoxic CD8 T cells and cancer cells) (*Dieckmann et al., 2016*), which exemplifies the intricate interplay between retrograde transport and cell polarity. Despite its importance, the connection between specific CIE mechanisms and retrograde transport in polarized cellular contexts remains largely underexplored.

Endophilin A (EndoA) proteins are key players in endocytosis, belonging to the BAR (Bin/Amphiphysin/Rvs) domain protein family known for their ability to sense and/or induce membrane curvature (*Peter et al., 2004*; *Gallop et al., 2006*). Mammalian cells express three isoforms of EndoA: EndoA1, A2, and A3 (*Sparks et al., 1996*; *Micheva et al., 1997*). EndoA1 is predominantly expressed in the brain, EndoA3 is abundant in both the brain and testes, and EndoA2 is ubiquitously expressed across tissues (*Giachino et al., 1997*; *So et al., 2000*). Initially, EndoA proteins were implicated in CME in neuronal cells, where they recruit dynamins and synaptojanins via interactions between their Src-homology 3 (SH3) domains and proline-rich domains (PRDs), facilitating vesicle scission (*Ringstad et al., 1997*; *Simpson et al., 1999*) and subsequent clathrin uncoating (*Gad et al., 2000*; *Milosevic et al., 2011*).

Recent studies have identified EndoA2 protein as a central player in clathrin-independent Fast Endophilin-Mediated Endocytosis (FEME) (*Boucrot et al., 2015*). EndoA2 has also been implicated in membrane scission events during the clathrin-independent uptake of Shiga and cholera toxins (*Renard et al., 2015*; *Simunovic et al., 2017*). Intriguingly, the three EndoA isoforms are not fully functionally redundant in CIE. Only EndoA3, and not EndoA1 or A2, mediates the endocytosis of the immunoglobulin (Ig)-like protein ALCAM (Activated Leukocyte Cell Adhesion Molecule). The endocytosis of ALCAM, the first identified EndoA3-specific cargo, operates in an EndoA3-dependent manner and is driven by extracellular Galectin-8 and glycosphingolipids, in agreement with the glycolipid-lectin (GL-Lect) endocytosis mechanism (*Renard et al., 2020*; *Johannes et al., 2016*; *Lakshminarayan et al., 2014*). Another Ig-like protein, L1 Cell Adhesion Molecule (L1CAM, alternative name CD171), has since been confirmed as a cargo for EndoA3-mediated CIE (*Lemaigre et al., 2023*).

ALCAM (CD166) is an adhesion molecule broadly expressed across tissues and overexpressed in various cancers, including bladder, prostate, and colorectal carcinomas (*Arai et al., 2002*; *Ferragut et al., 2021*; *Weichert et al., 2004*). ALCAM mediates cell–cell interactions either through homophilic binding with other ALCAM molecules (*van Kempen et al., 2001*) or by heterophilic binding with CD6 (*Bowen et al., 1995*). CD6, a member of the scavenger receptor cysteine-rich (SRCR) protein family, is predominantly expressed in immune cells, particularly T lymphocytes, and in certain neuronal cells (*Kamoun et al., 1981*; *Aruffo et al., 1997*). The ALCAM-CD6 interaction has been shown to play a critical role in immune synapse formation, providing essential co-stimulatory signals for optimal T cell activation and proliferation (*Zimmerman et al., 2006*; *Hassan et al., 2004*; *Gimferrer et al., 2004*).

Given the polarized nature of the immune synapse, we hypothesized that EndoA3-mediated endocytosis in cancer cells could influence immune synapse formation through the subsequent retrograde transport and polarized redistribution of its components, such as ALCAM. Moreover, since both ALCAM and L1CAM are Ig-like proteins, we speculated that EndoA3-mediated endocytosis might preferentially facilitate the uptake of Ig-like proteins. Interestingly, Intercellular Adhesion Molecule 1 (ICAM1, alternative name CD54), a well-characterized immune synapse component critical for co-stimulating T cell activation via its interaction with Lymphocyte Function-associated Antigen 1 (LFA-1) on the T cells (*Van Seventer et al., 1990*; *Zuckerman et al., 1998*; *Deeths and Mescher, 1999*), is also an Ig-like protein. This prompted us to investigate whether ICAM1 serves as an EndoA3-dependent CIE cargo in cancer cells and whether its role in immune synapse formation relies on EndoA3-mediated CIE and retrograde transport.

Through a combination of cell biology and immunology experiments, we identify ICAM1 as a novel EndoA3-dependent endocytic cargo. We further show that both ALCAM and ICAM1 undergo retromer-dependent retrograde transport to the TGN following EndoA3-mediated internalization. Disruption of EndoA3-mediated endocytosis or retromer-dependent retrograde transport in cancer cells significantly affects the response of autologous cytotoxic CD8 T cells, leading to reduced cytokine production and secretion, while paradoxically enhancing their lytic activity. Because ICAM1 reaches the immune synapse through polarized vesicular transport, we propose that this phenotype arises from defective polarized redistribution of immune synapse components, including ICAM1, from the Golgi to the plasma membrane. Consistent with this hypothesis, we find that disrupting EndoA3-mediated endocytosis in cancer cells reduces ICAM1 recruitment to the immune synapse, likely reflecting compromised structural integrity and formation. Interestingly, depleting EndoA3 or VPS26A increases the size of immune synapses, which reduces ICAM1 density at the contact zone. This synapse enlargement may represent an attempt by T cells to compensate for the loss of correctly localized components. However, this adaptation is insufficient to fully sustain CD8 T cell cytokine production and secretion, suggesting impaired synapse maturation. Conversely, we observe enhanced lytic activity, suggesting that these enlarged immune synapses promote more rapid detachment and re-engagement of T cells, potentially facilitating 'serial killing'. Collectively, our findings reveal an unexplored functional link between EndoA3-mediated clathrin-independent endocytosis and retrograde transport, which cooperate in cancer cells to relocate immune synapse components via the Golgi, thereby fine-tuning the balance between cytotoxic T cell cytokine secretion and lytic activity.

## Results

### ALCAM and ICAM1 are cargo clients of retromer-dependent retrograde trafficking

ALCAM and ICAM1 are known components of immune synapses (*Zimmerman et al., 2006*; *Hassan et al., 2004*; *Gimferrer et al., 2004*; *Zuckerman et al., 1998*; *Deeths and Mescher, 1999*). Given the polarized nature of these structures and the previously reported connection between retrograde transport and cell polarity (*Shafaq-Zadah et al., 2016*; *Carpier et al., 2018*), we investigated whether ALCAM and ICAM1 serve as cargoes for retrograde transport. To address this question, we employed a SNAP-tag-based assay to study retrograde transport of cargoes from the plasma membrane to the Golgi apparatus (*Johannes and Shafaq-Zadah, 2013*; *Figure 1A*). Briefly, benzylguanine (BG)-labeled cargo-specific antibodies (IgG) were incubated with cells stably expressing the Golgi-resident galactosyltransferase protein (GalT) fused to GFP and SNAP-tag (GalT-GFP-SNAP). If the cargo follows the retrograde route after endocytosis, it will be delivered to the TGN along with the BG-labeled antibody. Upon reaching the TGN, BG covalently reacts with the SNAP-tag, generating a large complex (IgG-SNAP-GFP-GalT), which can be immunopurified and analyzed by western blot.

We used a HeLa cell line stably expressing the Golgi-resident SNAP-tag construct (HeLa GalT-GFP-SNAP), validated in earlier studies (*Shafaq-Zadah et al., 2016*; *Shafaq-Zadah et al., 2021*). Additionally, we generated a LB33-MEL cell line stably expressing the same Golgi-resident SNAP-tag construct (LB33-MEL GalT-GFP-SNAP, *Figure 1—figure supplement 1A and B*). Of note, the LB33-MEL melanoma cell line was derived from a melanoma patient (LB33) in 1988. A specific autologous CD8 T cell line targeting this melanoma, obtained from the same patient, enables in vitro reconstitution of immune synapses (*Lehmann et al., 1995*). First, we validated the SNAP-tag functionality in both cell

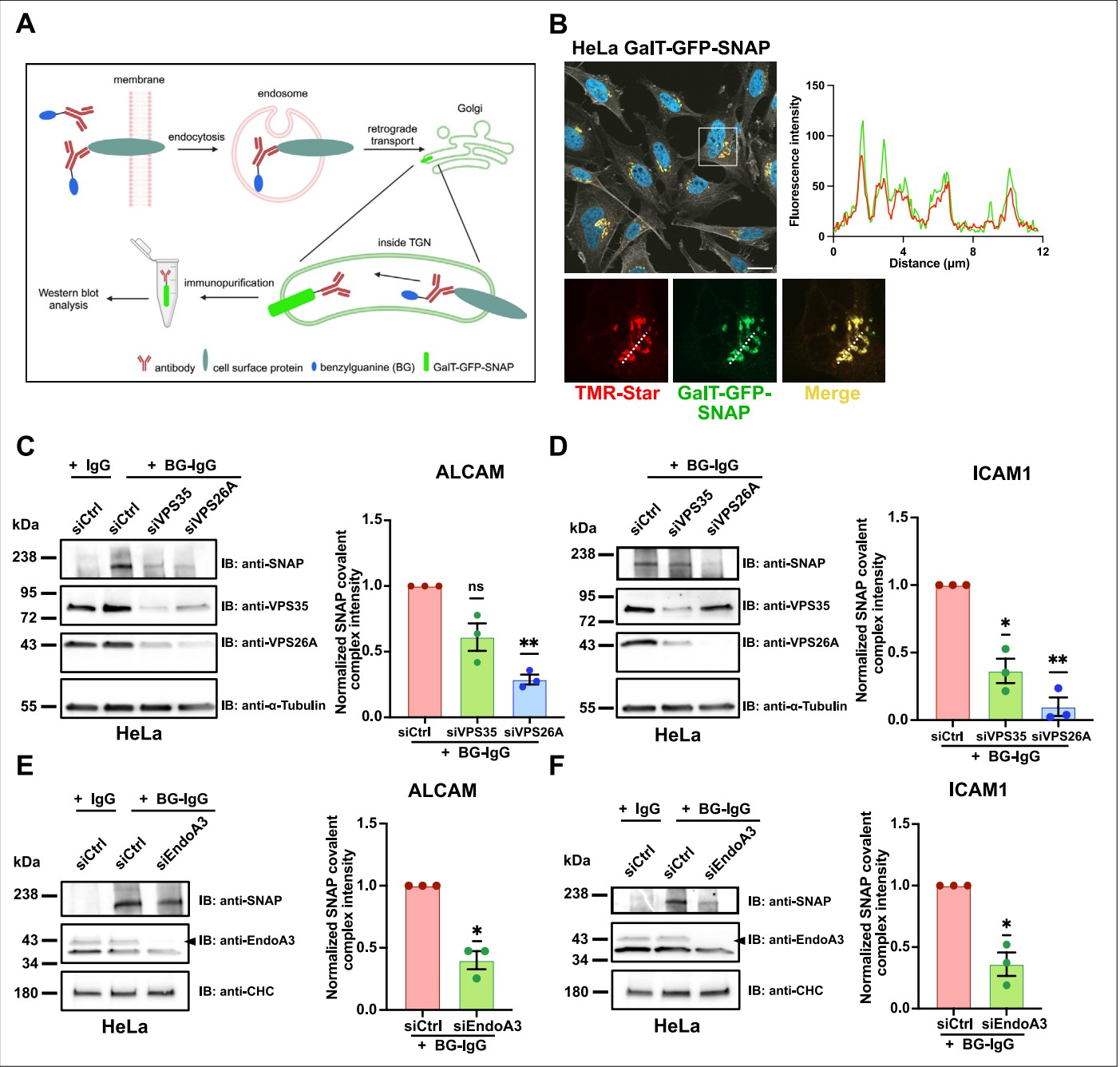

**Figure 1.** Immune synapse components ALCAM and ICAM1 are retrograde transport cargoes that rely on EndoA3 and retromer. (**A**) Illustration of the SNAP-tag-based BG-labeled antibody uptake assay to study membrane protein endocytosis and retrograde transport. (**B**) Confocal images of GalT-GFP-SNAP (green) and TMR-Star (red) in HeLa cells stably expressing the Golgi-resident GFP-fused SNAP-tag construct (HeLa GalT-GFP-SNAP). Actin (phalloidin, white) and nuclei (DAPI, blue) were also stained. Fluorescence intensity profile was made along the dashed line region in enlarged cropped area and shows the colocalization of both signals. Scale bar: 20 μm. (**C–F**) Retrograde transport of ALCAM and ICAM1. Continuous BG-labeled anti-ALCAM (**C, E**) and anti-ICAM1 (**D, F**) antibody uptake for 4 h at 37°C in HeLa GalT-GFP-SNAP cells. (**C, D**) Western blot analysis of HeLa GalT-GFP-SNAP cells transfected for 72 h with siRNAs: negative control (siCtrl) or against retromer subunits (siVPS35 and siVPS26A). Immunodetection made with anti-SNAP, anti-VPS35, anti-VPS26A, and anti-α-Tubulin (loading control) antibodies. Quantification of the covalent IgG-SNAP-GFP-GalT complex is shown as fractions of siCtrl condition (histogram). Quantification of VPS35 and VPS26A depletion is shown in *Figure 1—figure supplement 1C*. (**E, F**) Western blot analysis of HeLa GalT-GFP-SNAP cells transfected for 72 h with siRNAs: negative control (siCtrl) or against EndoA3 (siEndoA3). Immunodetection made with anti-SNAP, anti-EndoA3, and anti-clathrin heavy chain (CHC, loading control) antibodies. Quantification of the covalent IgG-SNAP-GFP-GalT complex (IB:anti-SNAP) is shown as fractions of siCtrl condition (histogram). Quantification of EndoA3 depletion is shown in *Figure 1—figure*

*Figure 1 continued on next page*

*Figure 1 continued*

**supplement 1H**. Data information: In (**B**), images are from a single experiment. Quantification data (**C–F**) are pooled from three independent experiments. Data are presented as mean ± SEM. *p<0.05, **p<0.01. One-sample *t* test and Wilcoxon test.

The online version of this article includes the following source data and figure supplement(s) for figure 1:

**Source data 1.** Original files for western blot analyses displayed in *Figure 1C–F*.

**Source data 2.** PDF files containing original western blots for *Figure 1C–F*.

**Figure supplement 1.** Immune synapse components ALCAM and ICAM1 are retrograde transport cargoes that rely on EndoA3, retromer, and Rab6.

**Figure supplement 1—source data 1.** Original files for western blot analyses displayed in *Figure 1—figure supplement 1D, F and I*.

**Figure supplement 1—source data 2.** PDF files containing original western blots for *Figure 1–figure supplement 1D, F and I*.

lines using the SNAP-Cell TMR-Star reagent (*Figure 1B* for HeLa; *Figure 1—figure supplement 1A* for LB33-MEL). We observed a strong and specific colocalization of TMR-Star with the GFP signal of the SNAP-tag constructs in both cell lines. Second, immunofluorescence experiments showed that the SNAP-tag construct co-distributes with the *TGN*-specific marker TGN46 in LB33-MEL cells, confirming its correct and specific localization (*Figure 1—figure supplement 1B*).

The retromer complex, composed of VPS26, VPS29, and VPS35 subunits in mammalian cells, is the best-studied machinery for retrograde transport (*Seaman, 2004*; *Hierro et al., 2007*). Two paralogues of VPS26 subunit exist: VPS26A, expressed in most tissues, and VPS26B, mainly enriched in the brain (*Bugarcic et al., 2011*; *Kerr et al., 2005*). To determine whether ALCAM and ICAM1 are cargoes of the canonical retromer complex, we conducted SNAP-tag-based retrograde transport assays in HeLa and LB33-MEL cell lines where VPS35 or VPS26A was knocked down by RNA interference (*Figure 1C and D* for HeLa; *Figure 1—figure supplement 1D and E* for LB33-MEL). BG-labeled anti-ALCAM or anti-ICAM1 antibodies were added to the cell culture medium for a 4 h continuous uptake and the production of a covalent IgG-SNAP-GFP-GalT complex was monitored by western blot (simplified as SNAP signal).

We observed that both ALCAM and ICAM1 are retrograde transport cargoes in cells treated with negative control siRNA, as IgG-SNAP-GFP-GalT complexes were detected by western blot after incubation with BG-labeled antibodies (band ~210 kDa; *Figure 1C and D* for HeLa; *Figure 1—figure supplement 1D* for LB33-MEL). As a negative control, unlabeled antibodies (IgGs) showed no detectable SNAP signal in western blot (*Figure 1C*). Interestingly, depletion of VPS35 or VPS26A caused a significant reduction in the SNAP signal for both cargoes (*Figure 1C and D*, *Figure 1—figure supplement 1C* for HeLa; *Figure 1—figure supplement 1D and E* for LB33-MEL). Of note, VPS26A depletion resulted in a stronger decrease in retrograde transport efficiency compared to VPS35 (*Figure 1C and D*).

To explore additional factors involved in retrograde transport, we examined the role of Rab6 GTPase, a critical molecular player in secretion which controls the formation of membrane tubules originating from the TGN (*Fourriere et al., 2019*; *Mallard et al., 2002*). Disrupting Rab6 indirectly inhibits retrograde transport to the TGN of cargoes requiring subsequent polarized anterograde redistribution (*Shafaq-Zadah et al., 2016*). Interestingly, Rab6 depletion impaired retrograde transport of ALCAM to the TGN, showing effects similar to retromer component depletion (*Figure 1—figure supplement 1F and G*).

Together, these data demonstrate that retromer-mediated retrograde transport is critical for trafficking ALCAM and ICAM1 to the Golgi and that this process requires efficient secretion from the TGN (as evidenced by the involvement of Rab6).

Importantly, previous studies from our lab demonstrated that the endocytosis of ALCAM is mediated by EndoA3-dependent CIE (*Renard et al., 2020*; *Tyckaert et al., 2022*). We therefore investigated whether EndoA3 depletion could affect the post-endocytic retrograde transport of ALCAM to the TGN. To test this, we depleted EndoA3 in HeLa cells using RNA interference (*Figure 1E*, *Figure 1—figure supplement 1H*) and observed 60% of reduction in the IgG-SNAP signal intensity (*Figure 1E*). Additionally, we performed retrograde transport assays with BG-labeled anti-ALCAM antibodies in the SNAP-tag-expressing LB33-MEL cell line (*Figure 1—figure supplement 1I*). Since LB33-MEL cells do not express detectable levels of EndoA3 as confirmed by western blot (*Renard et al., 2020*), we transiently transfected these cells with an EndoA3-GFP expression plasmid. Remarkably, EndoA3 expression increased the SNAP signal by ~50% compared to the control condition

where cells were transfected with a plasmid expressing free GFP (*Figure 1—figure supplement 1I*). Given that ICAM1 is also a member of the Ig-like protein family, we investigated whether the retrograde transport of this cargo depends on EndoA3. Interestingly, EndoA3 depletion in HeLa cells resulted in a significant reduction of the ICAM1 IgG-SNAP signal by 64% (*Figure 1F*), comparable to the effect observed for ALCAM (*Figure 1E*).

Collectively, these results suggest that (i) ICAM1 is potentially a novel EndoA3-dependent cargo, (ii) substantial fractions of ALCAM and ICAM1 are transported to the TGN in a retromer-dependent manner after EndoA3-mediated CIE, and (iii) introducing EndoA3 in EndoA3-negative cells enhances the retrograde transport of these cell adhesion molecules to the TGN. These findings underscore a tight functional link between a specific clathrin-independent endocytic mechanism – mediated by EndoA3 – and the subsequent retrograde transport of cargoes to the TGN.

## EndoA3 expression enhances the internalization of immune synapse components ALCAM and ICAM1 in cancer cells

To further confirm that ICAM1 is an EndoA3-dependent cargo, we employed live-cell TIRF imaging to directly observe the dynamics of EndoA3 and ICAM1 at the plasma membrane of both HeLa and LB33-MEL cell lines stably expressing EndoA3-GFP. Of note, the EndoA3-GFP-expressing HeLa cell line was generated and validated in earlier studies from our lab (*Renard et al., 2020*; *Lemaigre et al., 2023*), while the EndoA3-GFP-expressing LB33-MEL cell line (hereafter referred to as LB33-MEL EndoA3+) was developed specifically for this study. Both cell lines were then transiently transfected to express ICAM1-mScarlet.

Interestingly, EndoA3-GFP formed circular structures around ICAM1 patches in both cell lines (*Figure 2A and B*, *Figure 2—figure supplement 1A and B*, white arrows; *Figure 2—videos 1, 2, 5 and 6*), and ICAM1 signals gradually disappeared from the plasma membrane (*Figure 2A and B*, *Figure 2—figure supplement 1A and B*; *Figure 2—videos 1, 2, 5 and 6*). In addition, ICAM1-positive punctate structures were observed, where EndoA3-GFP was recruited before both protein signals disappeared from the cell surface (*Figure 2A and B*, *Figure 2—figure supplement 1B*, kymographs; *Figure 2—videos 3, 4 and 7*), indicating active endocytic events. This behavior mirrors TIRF observations previously reported for other EndoA3-dependent endocytic cargoes, such as ALCAM and L1CAM (*Renard et al., 2020*; *Lemaigre et al., 2023*). Collectively, these observations confirm that ICAM1 is a novel EndoA3-dependent endocytic cargo.

To investigate the role of EndoA3-dependent CIE and subsequent retrograde transport in cancer cells on the response of cytotoxic CD8 T lymphocytes, we further used the LB33-MEL cellular model. This cell line expresses a mutated antigenic peptide encoded by the *MUM3* gene. The MUM-3 peptide is presented on the cell surface by HLA-A*68012 molecules, an HLA-A28 subtype specific to patient LB33 (*Lehmann et al., 1995*; *Baurain et al., 2000*). A CD8 T cell line targeting the MUM-3 peptide, derived from the same patient in 1990 (*Lehmann et al., 1995*), can form immune synapses with LB33-MEL cells in vitro. Although LB33-MEL cells lack detectable levels of EndoA3 (*Renard et al., 2020*), we have observed above that exogenous expression of EndoA3 enhances the retrograde transport of cell adhesion molecules (*Figure 1—figure supplement 1I*).

Consistent with this result, high-resolution Airyscan confocal images showed a clear colocalization between EndoA3-GFP and ALCAM, after a 15-min uptake of anti-ALCAM antibodies in LB33-MEL EndoA3+ cells (*Figure 2C*, white arrowheads). This observation confirmed further that this cell line recapitulates EndoA3-mediated endocytosis. Additional quantitative analyses revealed that the uptake of anti-ALCAM antibodies was three times higher in LB33-MEL EndoA3+ cells compared to wild-type or empty plasmid-transfected LB33-MEL cells (*Figure 2D*, *Figure 2—figure supplement 2A*). In addition, we performed a reverse-rescue uptake assay in LB33-MEL EndoA3+ cells transfected with EndoA3 siRNAs (*Figure 2E*, *Figure 2—figure supplement 2B*). ALCAM uptake was significantly reduced by 55% upon EndoA3 knockdown (*Figure 2E*). Of note, exogenously expressed EndoA3-GFP protein was efficiently knocked down by siRNAs in LB33-MEL EndoA3+ cells, with an 80% reduction as shown by immunoblotting (*Figure 2—figure supplement 2C*).

Taken together, these findings confirm that LB33-MEL EndoA3+ cells recapitulate functional EndoA3-dependent CIE, enhancing the uptake of canonical EndoA3-dependent cargoes. This cell line provides a robust model to assess the effects of EndoA3-mediated CIE and subsequent retrograde transport on CD8 T cell response.

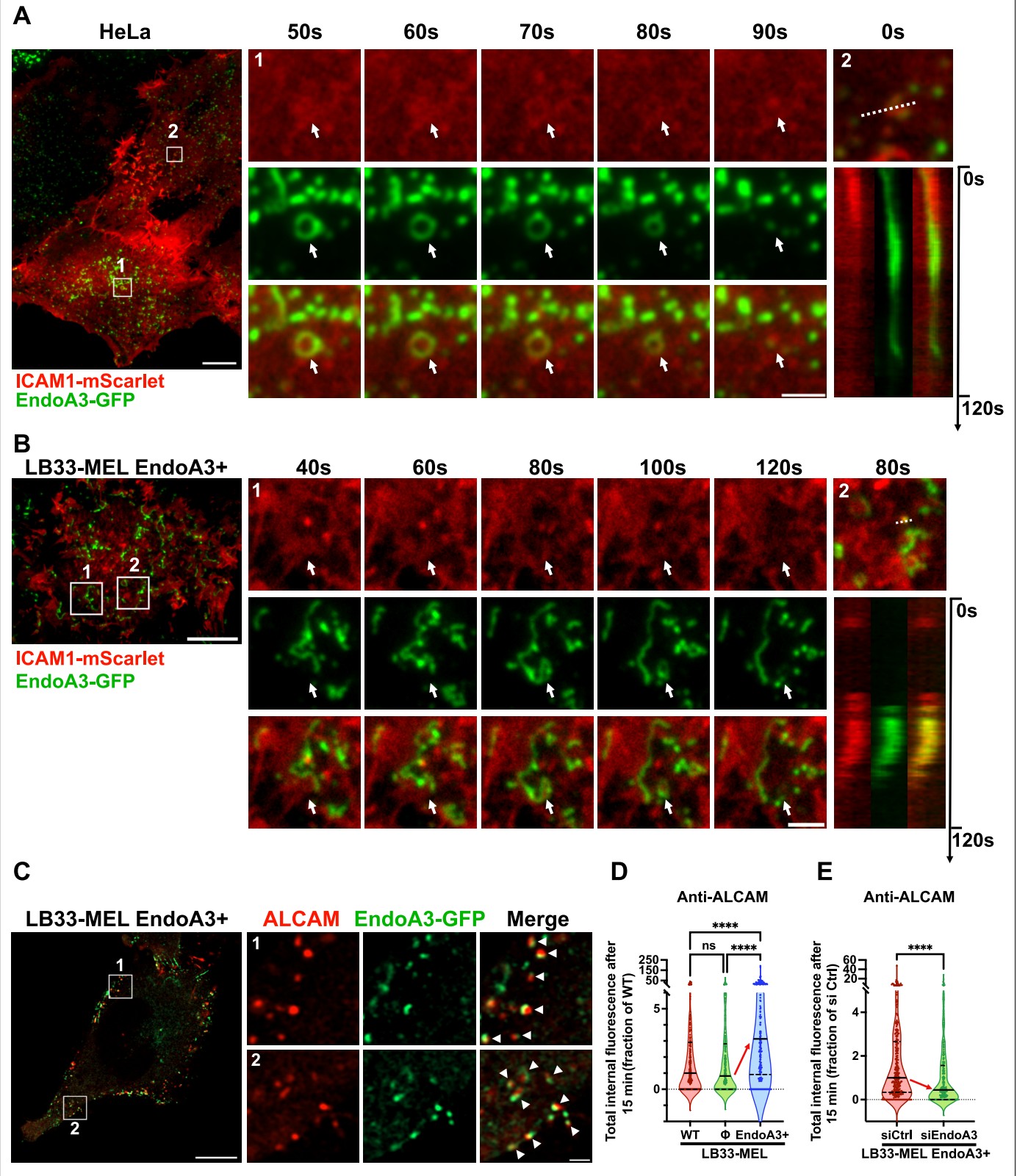

**Figure 2.** EndoA3-dependent CIE mediates the uptake of immune synapse components ALCAM and ICAM1 in cancer cells. (**A, B**) Live-cell TIRF images of EndoA3-GFP (stable) and ICAM1-mScarlet (transient) in HeLa (**A**) and LB33-MEL (**B**) cells. Time series show enlarged cropped areas corresponding to region 1 in the full-size images and are extracted from *Figure 2—videos 1 and 2*. White arrows indicate dynamic co-distribution of both signals. Kymographs were made along the dashed lines in the enlarged cropped areas corresponding to region 2 (**A, B**; *Figure 2—videos 3 and 4*). Scale bars:

*Figure 2 continued on next page*

*Figure 2 continued*

10 μm (full-size image) and 2 μm (enlarged cropped areas). (**C–E**) Continuous uptake of anti-ALCAM antibody for 15 min at 37°C in the following LB33-MEL cell lines: wild-type (WT, **D**), stably transfected with empty plasmid (Φ, **D**), or stably expressing EndoA3-GFP (LB33-MEL EndoA3+, **C–E**). In (**E**), cells were transfected with siRNAs: negative control (siCtrl) or against EndoA3 (siEndoA3). Quantification of EndoA3 depletion by western blots in *Figure 2— figure supplement 2C*. (**C**) Airyscan images of Anti-ALCAM (red) and EndoA3-GFP (green). White arrowheads show colocalization between ALCAM and EndoA3. Scale bars: 10 μm (full-size image), 1 μm (enlarged cropped areas). (**D, E**) Quantifications of anti-ALCAM internalization, expressed as fractions of WT condition (**D**) or siCtrl condition (**E**). (**D**) n cells: WT, n=270; Φ, n=279; EndoA3+, n=274. (**E**) n cells: siCtrl, n=350; siEndoA3, n=234. Representative image examples in *Figure 2—figure supplement 2A and B*. Data information: All images (**A–C**) are representative of two independent experiments. In (**D, E**), data are pooled from three independent experiments. Data are presented as median and quartiles. ns, not significant. ****p<0.0001 (**D**, Kruskal–Wallis test with Dunn's multiple comparison test; **E**, Mann–Whitney test).

The online version of this article includes the following video, source data, and figure supplement(s) for figure 2:

**Figure supplement 1.** ICAM1 is an EndoA3-dependent endocytic cargo in cancer cells.

**Figure supplement 2.** EndoA3-dependent CIE mediates the uptake of immune synapse components ALCAM in LB33-MEL cells.

**Figure supplement 2—source data 1.** Original files for western blot analysis displayed in *Figure 2–figure supplement 2C*.

**Figure supplement 2—source data 2.** PDF file containing original western blots for *Figure 2–figure supplement 2C*.

**Figure 2—video 1.** Gradual disappearance of EndoA3-enclosed ICAM1 from the plasma membrane of a HeLa cell, related to *Figure 2A*, cropped area 1.
https://elifesciences.org/articles/105821/figures#fig2video1

**Figure 2—video 2.** Gradual disappearance of EndoA3-enclosed ICAM1 from the plasma membrane of an LB33-MEL cell, related to *Figure 2B*, cropped area 1.
https://elifesciences.org/articles/105821/figures#fig2video2

**Figure 2—video 3.** A punctate ICAM1-positive structure recruits EndoA3, and both signals disappear simultaneously from the plasma membrane of HeLa cells, related to *Figure 2A*, kymograph of cropped area 2.
https://elifesciences.org/articles/105821/figures#fig2video3

**Figure 2—video 4.** A punctate ICAM1-positive structure recruits EndoA3, and both signals disappear simultaneously from the plasma membrane of an LB33-MEL cell, related to *Figure 2B*, kymograph of cropped area 2.
https://elifesciences.org/articles/105821/figures#fig2video4

**Figure 2—video 5.** Dynamic ICAM1-positive patch, bordered by 'lasso-like' EndoA3 structures, progressively shrinks and disappears from the plasma membrane of a HeLa cell, related to *Figure 2–figure supplement 1A*.
https://elifesciences.org/articles/105821/figures#fig2video5

**Figure 2—video 6.** Dynamic ICAM1-positive patch, bordered by 'lasso-like' EndoA3 structures, progressively shrinks and disappears from the plasma membrane of an LB33-MEL cell, related to *Figure 2–figure supplement 1B*, cropped area 1.
https://elifesciences.org/articles/105821/figures#fig2video6

**Figure 2—video 7.** A punctate ICAM1-positive structure recruits EndoA3, and both signals disappear simultaneously from the plasma membrane of an LB33-MEL cell, related to *Figure 2–figure supplement 1B*, kymograph of cropped area 2.
https://elifesciences.org/articles/105821/figures#fig2video7

## EndoA3-dependent CIE and retromer-mediated retrograde transport in cancer cells modulate cytotoxic CD8 T cell cytokine secretion and lytic activity

Based on our results and considering that ALCAM and ICAM1 are key components of immune synapses, we hypothesized that their clathrin-independent endocytosis and subsequent retrograde transport might influence CD8 T cell response. To test this hypothesis, we conducted CD8 T cell stimulation assays using LB33-MEL cells, in the presence or absence of EndoA3. Specifically, LB33-MEL EndoA3+ cells were treated with either control or EndoA3-targeting siRNAs before co-culture with CD8 T cells. Intracellular cytokine production in T cells was then assessed by flow cytometry (*Figure 3A*).

When CD8 T cells were stimulated with EndoA3-depleted LB33-MEL cells, we observed a significant decrease in both the percentage of T cells producing interleukin-2 (IL-2) and tumor necrosis factor α (TNFα) (*Figure 3B*, upper and middle panels), as well as the absolute amounts of these cytokines (*Figure 3C*, upper and middle panels), compared to CD8 T cells stimulated with LB33-MEL EndoA3+ cells. Although the percentage of T cells producing interferon γ (IFNγ) remained unchanged (*Figure 3B*, bottom panel), the absolute amount of IFNγ produced was still significantly reduced

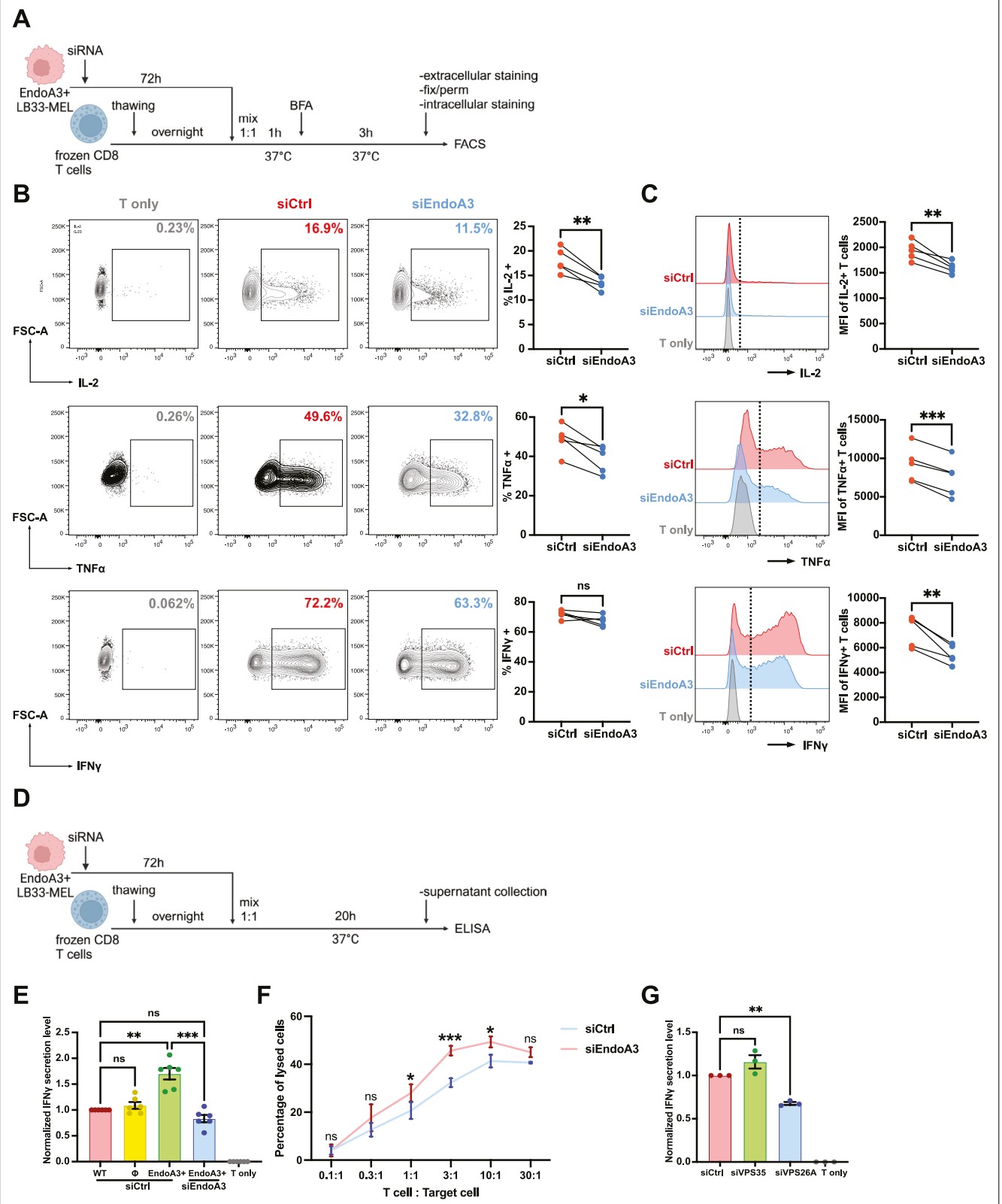

**Figure 3.** Inhibition of EndoA3-dependent endocytosis and retrograde transport impairs T cell activation but increases their lytic activity. (**A**) Scheme of flow cytometry analysis of cytokine production inside CD8 T cells stimulated by siRNA-transfected LB33-MEL EndoA3+ cells. BFA, Brefeldin A. (**B, C**) Flow cytometry analysis of CD8 T cell intracellular cytokine production after co-culture for 4 h at 37°C with: no LB33-MEL (T only), LB33-MEL EndoA3+ transfected with negative control siRNA (siCtrl) or with siRNA against EndoA3 (siEndoA3). Quantification of EndoA3 depletion is shown in *Figure 3—*

*Figure 3 continued on next page*

*Figure 3 continued*

**figure supplement 1A**. (**B**) Representative examples and quantifications (scatter plots) of the percentages of cytokine producing CD8 T cells (up, IL-2; middle, TNFα; bottom, IFNγ) after being stimulated by siCtrl or siEndoA3 treated LB33-MEL EndoA3+ cells. (**C**) Representative examples and quantifications (scatter plots) of the absolute amount of cytokines produced by CD8 T cells (up, IL-2; middle, TNFα; bottom, IFNγ; presented as median fluorescence intensity, MFI) after stimulation by siCtrl or siEndoA3 treated LB33-MEL EndoA3+ cells. (**D**) Scheme of ELISA analysis of cytokine secretion from CD8 T cells stimulated with siRNA transfected LB33-MEL EndoA3+ cells. (**E**) Quantification of IFNγ secretion (detected by ELISA) from CD8 T cells, cultured alone (T only) or co-cultured for 20 h with the following LB33-MEL cell lines: wild-type LB33-MEL cells (WT), LB33-MEL cells stably transfected with empty (Φ) or EndoA3-GFP encoding plasmid (EndoA3+), treated with negative control siRNA (siCtrl) or with EndoA3-targeting siRNA (siEndoA3). Data are presented as fractions of WT siCtrl condition. The absolute concentration of secreted IFNγ in the supernatant of WT siCtrl condition is 2203±201 (mean ± SEM) pg/mL. (**F**) Quantification of CD8 T cell cytolytic efficiency against LB33-MEL EndoA3+ cells transfected with negative control siRNA (siCtrl) or siRNA against EndoA3 (siEndoA3). CD8 T cell killing efficiency was determined by Chrome 51 release assay and is presented as percentage of lysed LB33-MEL EndoA3+ cells. Different T cell:Target cell ratios were tested. (**G**) Quantification of IFNγ secretion (detected by ELISA) from CD8 T cells, cultured alone (T only) or co-cultured for 20 h at 37°C with EndoA3+LB33 MEL cells transfected with siRNAs: negative control (siCtrl), or against retromer subunits (siVPS35 or siVPS26A). Data are presented as fractions of siCtrl condition. The absolute concentration of secreted IFNγ in the supernatant of siCtrl condition is 4077±99.62 (mean ± SEM) pg/mL. Validation of VPS35 and VPS26A depletion is shown in ***Figure 3—figure supplement 1L***. Data information: In (**B, C**), data are pooled from five independent experiments. In (**E**), data are pooled from six independent experiments. In (**F, G**), data are pooled from three independent experiments. In (**E, G**), data are presented as mean ± SEM. ns, not significant. *p<0.05, **p<0.01, ***p<0.001 (**B** and **C**, paired *t* test; **E**, RM one-way ANOVA with Tukey's multiple comparison test; **F**, two-way ANOVA with Sidak's multiple comparison test; **G**, RM one-way ANOVA with Dunnett's multiple comparison test).

The online version of this article includes the following source data and figure supplement(s) for figure 3:

**Figure supplement 1.** Effect of EndoA3 depletion on morphological parameters and surface levels of several markers in LB33-MEL EndoA3+ cells or CD8 T cells.

**Figure supplement 1—source data 1.** Original files for western blot analysis displayed in ***Figure 3—figure supplement 1L***.

**Figure supplement 1—source data 2.** PDF file containing original western blots for ***Figure 3—figure supplement 1L***.

(***Figure 3C***, bottom panel). Of note, the efficiency of EndoA3-GFP depletion was confirmed by flow cytometry (***Figure 3—figure supplement 1A***).

Next, we assessed IFNγ secretion by CD8 T lymphocytes in co-culture supernatants using enzyme-linked immunosorbent assay (ELISA) (***Figure 3D***). Wild-type, empty plasmid control, and EndoA3-GFP-expressing LB33-MEL cells all stimulated IFNγ secretion by CD8 T cells, compared to the control condition where T cells were cultured alone (***Figure 3E***). Notably, EndoA3 expression significantly enhanced CD8 T cell response, leading to an ~70% higher IFNγ secretion compared to wild-type or empty plasmid–transfected cancer cells (***Figure 3E***). This stimulatory effect was lost upon EndoA3 depletion by RNA interference (***Figure 3E***). To exclude the possibility that these effects are due to alterations in the biophysical properties of cancer cells following siRNA treatment (i.e., membrane tension), we analyzed their spreading area (***Figure 3—figure supplement 1B***), aspect ratio (***Figure 3—figure supplement 1C***), and roundness (***Figure 3—figure supplement 1D***). None, or only minimal, changes were detected, indirectly indicating that the observed differences in T cell activation were likely not attributable to variations in cancer cell stiffness. In addition, to exclude the possibility that these effects are due to drastic changes of ICAM1 and ALCAM abundance at the plasma membrane, we measured their surface level by flow cytometry in cancer cells expressing or depleted of EndoA3 (***Figure 3—figure supplement 1E and F***). We observed only minor variations which cannot account for the very significant drop in IFNγ secretion upon EndoA3 depletion.

To further dissect the mechanisms underlying EndoA3-mediated modulation of T cell responses, we examined additional parameters of CD8 T cell activation. We first analyzed surface expression of activation markers, including PD-1, CD137, and Tim-3, on CD8 T cells co-cultured with EndoA3-expressing or -depleted LB33-MEL cells (***Figure 3—figure supplement 1G–I***). None of these markers were significantly altered by EndoA3 knockdown. Similarly, EndoA3 depletion did not affect T cell proliferation (***Figure 3—figure supplement 1J***) nor degranulation (***Figure 3—figure supplement 1K***). Unexpectedly, chromium release assays revealed that CD8 T cells exhibited increased cytotoxic activity when co-cultured with EndoA3-depleted cancer cells, approximately 15% higher at a 3:1T cell-to-target cell ratio, corresponding to the maximal effect observed (***Figure 3F***). Although seemingly paradoxical, this finding may align with a model in which EndoA3 depletion leads to disturbed immune synapse organization, thereby promoting shorter contacts with CD8 T cells and enhancing serial killing capacity. Shorter contacts with cancer cells may also explain the concomitant decrease in IFNγ secretion under the same conditions (see Discussion).

Together with our previous results, these observations support the hypothesis that EndoA3-mediated CIE in cancer cells influences retrograde transport and the subsequent polarized redistribution of immune synapse components, such as ALCAM and ICAM1, thereby affecting T cell response. To further test this model, we disrupted the retromer complex in LB33-MEL EndoA3+ cells by depleting the subunits VPS26A and VPS35 using RNA interference and assessed the effects on CD8 T cell activation by measuring cytokine secretion. While VPS35 depletion had no significant impact, loss of VPS26A markedly impaired IFNγ secretion (*Figure 3G*, *Figure 3—figure supplement 1L*).

In summary, our data demonstrate that EndoA3-dependent endocytosis in cancer cells contributes to CD8 T cell response by promoting retrograde transport of immune synapse components such as ALCAM and ICAM1. This process likely enables their polarized redistribution from the TGN to the cell surface, facilitating efficient immune synapse assembly. Conversely, EndoA3 depletion in target cells reduces CD8 T cell cytokine production and secretion but increases their lytic activity, reflecting changes in immune synapse organization and potentially enhanced serial killing dynamics driven by more transient immune synapse interactions.

## Inhibition of EndoA3-dependent endocytosis and retrograde transport affects ICAM1 recruitment in the vicinity of the contact zone with CD8 T cells

Building on the data presented above, effective EndoA3-dependent endocytosis tunes CD8 T cell response, possibly through an increased retrograde flux to the TGN and the subsequent polarized redistribution of immune synapse components to the plasma membrane. According to this hypothesis, depletion of EndoA3 or retromer subunits should perturb the recruitment of components to the immune synapse. Consequently, the structural features of the immune synapses would likely be affected. To test this, we monitored the recruitment of ICAM1 to the immune synapse using spinning-disk and high-resolution Airyscan confocal microscopy, given the well-established role of ICAM1 in T cell activation (*Van Seventer et al., 1990*).

While we were able to transiently transfect LB33-MEL cells with an ICAM1-expressing plasmid (*Figure 2B*), the transfection efficiency was low, thereby limiting the possibility of using LB33-MEL cells for immune synapse imaging. To overcome this, we generated a stable HLA-A*68012-expressing HeLa cell line. Wild-type HeLa cells lack HLA-A*68012 and are therefore unable to present the MUM-3 antigenic peptide to stimulate CD8 T cell activation (*Figure 4—figure supplement 1A*). Upon addition of exogenous MUM-3 peptide, the HLA-A*68012-expressing HeLa cells successfully stimulated anti-MUM-3 CD8 T cells to secrete IFNγ (*Figure 4—figure supplement 1A*) and proliferate (*Figure 4—figure supplement 1B*). Thereby, these data confirmed the functionality of this HLA-A*68012-expressing HeLa cell line.

To monitor the recruitment of ICAM1 to the immune synapse, HLA-A*68012-expressing HeLa cells transiently expressing EGFP-tagged ICAM1 were loaded with MUM-3 peptide and co-cultured with CD8 T cells. Using high-speed spinning-disk confocal microscopy, we observed a flux of ICAM1-positive tubulo-vesicular carriers emerging from the perinuclear region (corresponding to the Golgi area) and moving toward the contact site with the CD8 T cell (*Figure 4A*; *Figure 4—video 1*), with fusion events of carriers occurring at the developing immune synapse-like conjugate (*Figure 4B*; *Figure 4—video 2*). AI-based segmentation and tracking analyses showed that ICAM1-positive carrier trajectories were predominantly oriented toward the forming immune synapse (*Figure 4C*), whereas carriers moving toward other cellular regions were markedly less frequent (*Figure 4D and E*), as revealed by the quantification of track densities (*Figure 4F*) and track proportions in the different directions (*Figure 4G*) in these different regions. Aligning with these observations, immunofluorescence experiments using HeLa cells transiently expressing ICAM1-mScarlet revealed a strong accumulation of ICAM1 signal in the contact zone with CD8 T cells (*Figure 4—figure supplement 1C*, white arrowheads). Altogether, these results provide direct evidence for polarized ICAM1 transport via vesicular trafficking toward the immune synapse.

Next, we treated ICAM1-mScarlet-expressing HeLa cells with control or EndoA3 siRNAs and incubated them with CD8 T cells (*Figure 5A and B*, *Figure 5—figure supplement 1A and B*). To facilitate quantification, this experiment was conducted on cells in suspension, which adopt a regular spherical shape, making them easier to segment using quantification algorithms (*Figure 5A*, *Figure 5—figure supplement 1A*, phalloidin channel). We measured the ICAM1 recruitment ratio in the vicinity of T

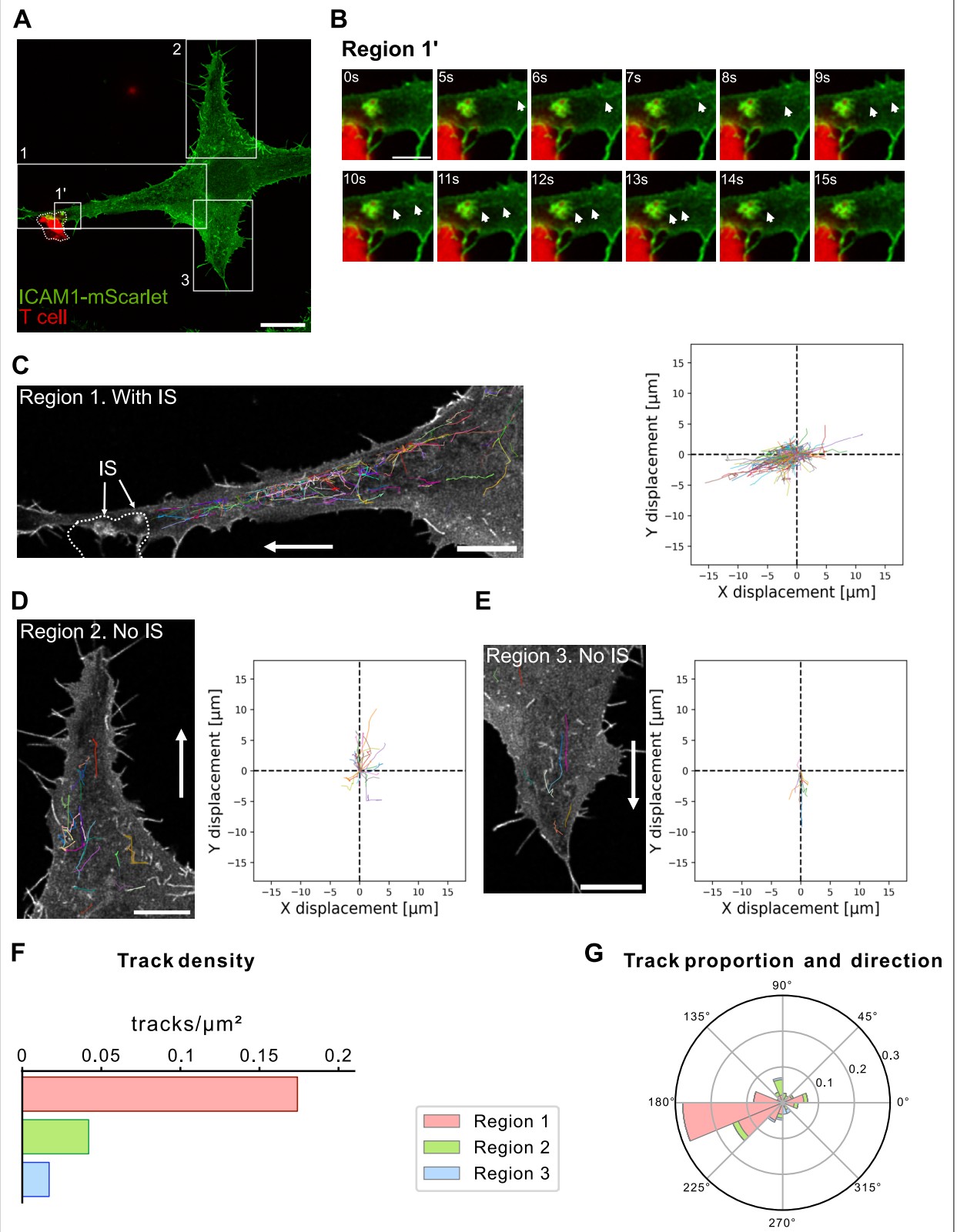

**Figure 4.** ICAM1 is directionally transported to the immune synapse via vesicular trafficking in cancer cells. (**A**) Representative live-cell spinning-disk confocal image of an immune synapse-like conjugate formed between a CD8 T cell (red) and an adherent, stable HLA-A*68012–expressing HeLa cell transiently expressing ICAM1-mScarlet (green) (*Figure 4—video 1*). Three regions were selected for quantitative analysis: region 1 at the T cell contact site, and regions 2 and 3 away from the conjugate (quantified in panels **C–G**). The T cell is delineated by a dotted line. Scale bar, 20 μm. (**B**) Time series

*Figure 4 continued on next page*

*Figure 4 continued*

of the enlarged cropped area (region 1') from panel (**A**), extracted from *Figure 4—video 1* and shown in *Figure 4—video 2*. White arrowheads indicate ICAM1-positive tubulo-vesicular carriers moving toward the contact area and fusing with the conjugate membrane. Scale bar, 5 µm. (**C–E**) Tracking of ICAM1-positive carriers in the three regions defined in panel (**A**)-region 1 (**C**, with immune synapse) and regions 2–3 (**D, E**, without immune synapse). For each region, all recorded tracks are displayed on the cell image (left) and in corresponding X–Y displacement plots (right). Scale bar, 10 µm. (**F**) Quantification of track density in each region. Track density is markedly higher in region 1 where the CD8 T cell contact is. (**G**) Quantification of track directionality and proportion across regions. A higher proportion of tracks is oriented toward the T cell contact site in region 1. Data information: Images and quantifications are from a single video, representative of two independent experiments. Number of tracks computed in each region: region 1, n = 149; region 2, n = 28; region 3, n = 9.

The online version of this article includes the following video and figure supplement(s) for figure 4:

**Figure supplement 1.** HLA-A*68012-expressing HeLa cells stimulate the activation of anti-MUM-3 CD8 T cells in the presence of MUM-3 peptide.

**Figure 4—video 1.** ICAM1 tubulo-vesicular carrier flux is stronger and directed toward the contact region of an immune synapse-like conjugate formed between a HeLa cell and a T cell, related to *Figure 4*.

https://elifesciences.org/articles/105821/figures#fig4video1

**Figure 4—video 2.** ICAM1 tubulo-vesicular carriers fuse at the ICAM1-enriched contact region of an immune synapse-like conjugate formed between a HeLa cell and a T cell, related to *Figure 4B* (region 1').

https://elifesciences.org/articles/105821/figures#fig4video2

---

cells (see Materials and Methods). Surprisingly, the ICAM1 signal was not confined to the contact region of the conjugates, but often expanded around the entire T cell (*Figure 5A*, *Figure 5—figure supplement 1A*). Quantifications revealed that depletion of EndoA3 in HLA-A*68012-expressing HeLa cells caused a slight but significant decrease in ICAM1 recruitment to the vicinity of anti-MUM-3 CD8 T cells (*Figure 5B*, *Figure 5—figure supplement 1B*), consistent with the previously observed impaired T cell cytokine secretion (*Figure 3*).

We then explored the consequences of EndoA3 or VPS26A depletion in cancer cells on the morphology of immune synapses. To study this, we returned to the LB33-MEL EndoA3+ cell line and imaged immune synapses formed with the CD8 T cells using Airyscan confocal microscopy (*Figure 5C and E*, *Figure 5—figure supplement 2A and B*). This experiment was also performed on cells in suspension for ease of quantitative analysis. We observed that immune synapses formed between CD8 T cells and EndoA3-depleted or VPS26A-depleted LB33-MEL EndoA3+ cells were larger than those formed between CD8 T cells and control LB33-MEL EndoA3+ cells. Quantification of immune synapse sizes confirmed this observation, showing that the immune synapses were 24% and 12% larger following EndoA3 and VPS26A depletion, respectively (*Figure 5D and F*). Of note, the efficiency of siRNA transfections was validated by western blot (*Figure 5—figure supplement 2C and D*).

Together, these results show that inhibiting CIE or retrograde transport in cancer cells reduces ICAM1 recruitment in the vicinity of the contact zones with CD8 T cells and alters the organization of immune synapses. The resulting synapses are enlarged and display lower ICAM1 density, confirming structural alterations that were indirectly suspected from earlier CD8 T cell activation experiments (*Figure 3*, *Figure 3—figure supplement 1*). Indeed, despite their increased size, these synapses fail to promote efficient cytokine secretion – which usually requires fully mature synapses – although higher lytic activity is observed (*Figure 3*, *Figure 3—figure supplement 1*).

In conclusion, disrupting endoA3-mediated CIE and downstream retrograde transport in cancer cells impairs the recruitment of immune synapse components, such as ICAM1, likely by altering their polarized redistribution from TGN to the plasma membrane. As a result, immune synapses may be less mature, with potential consequences on CD8 T cell responses. CD8 T cells may attempt to compensate for reduced recruitment of synaptic components by expanding their contact area with cancer cells, though this remains insufficient to support efficient cytokine production and secretion. This condition might also promote more transient contacts of CD8 T cells (*i.e.* faster detachment and re-engagement), which may favor serial killing (*Figure 5G*).

## Discussion

In this study, we discovered that ICAM1 is a new cargo of EndoA3-mediated CIE, expanding the list of known EndoA3-dependent cargoes, which previously included the Ig-like cell proteins ALCAM and L1CAM (*Renard et al., 2020*; *Lemaigre et al., 2023*). Interestingly, the first study by *Boucrot et al.,*

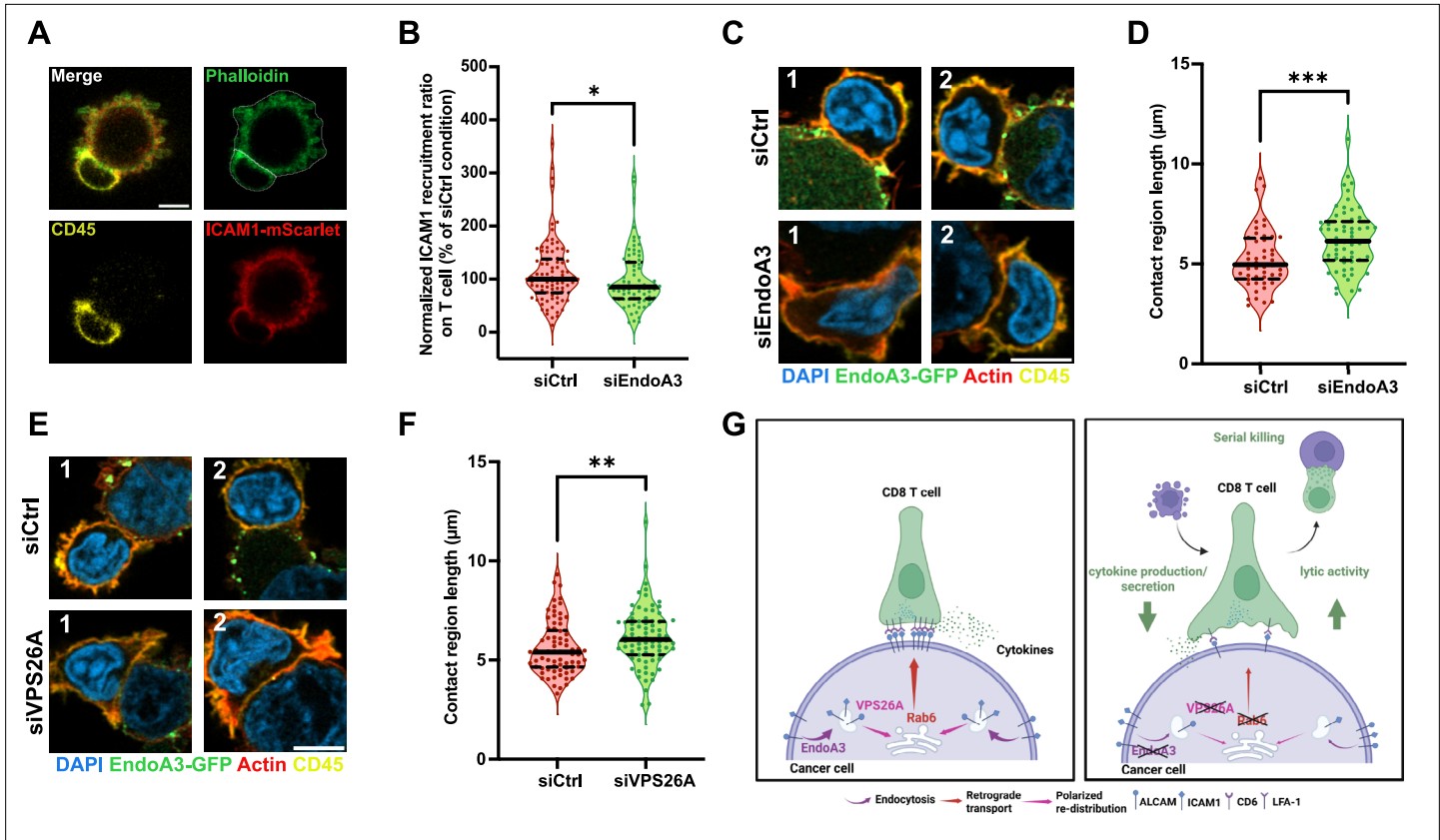

**Figure 5.** Inhibition of EndoA3-mediated endocytosis and retrograde transport affects ICAM1 recruitment to and structure of immune synapses. (**A**) Airyscan images of an immune synapse-like conjugate formed between a CD8 T cell (stained for CD45, yellow) and a stable HLA-A*68012-expressing HeLa cell (stained for actin, green) transiently expressing ICAM1-mScarlet (red) in suspension. Cell segmentation (white contour) was based on the actin staining for further quantifications (**B**). Scale bar: 5 µm. (**B**) Quantification of relative ICAM1 recruitment to the vicinity of CD8 T cell when an immune synapse-like conjugate is formed between a CD8 T cell and a stable HLA-A*68012-expressing HeLa cell transiently expressing ICAM1-mScarlet in suspension. HLA-A*68012-expressing HeLa cells were transfected for 72 h with siRNAs: negative control (siCtrl) or against EndoA3 (siEndoA3). Western blot analysis of EndoA3 depletion is shown in *Figure 5—figure supplement 1B*. n conjugates: siCtrl, n=85; siEndoA3, n=75. (**C, E**) Airyscan images of immune synapse-like conjugates formed between CD8 T cells (stained for CD45, yellow) and stable EndoA3-GFP-expressing LB33-MEL cells (EndoA3-GFP, green) transfected for 72 h with different siRNAs: negative control (siCtrl), siRNAs against EndoA3 (siEndoA3, **C**) or against VPS26A (siVPS26A, **E**). Actin (phalloidin, red) and nuclei (DAPI, blue) were also stained. Two examples are displayed per condition. Scale bar: 5 µm. (**D, F**) Quantifications of the sizes of immune synapse-like conjugates formed between CD8 T cells and stable EndoA3-GFP-expressing LB33-MEL cells transfected with different siRNAs from images in panels (**C**) and (**E**). Western blot analyses of EndoA3 and VPS26A depletion are shown in *Figure 5—figure supplement 2C and D*. (**D**) n conjugates: siCtrl, n=48; siEndoA3, n=58. (**F**) n conjugates: siCtrl, n=74; siVPS26A, n=78. (**G**) Working model. Immune synapse components ALCAM and ICAM1 are endocytosed into cancer cells through EndoA3-mediated CIE and are subsequently transported to the *TGN* in a retromer-dependent retrograde manner. Upon immune synapse formation, immune synapse components, such as ALCAM and ICAM1, are efficiently recruited to the contact area with CD8 T cells, likely from the *TGN* by polarized re-distribution, which can stabilize the immune synapse and promote cytokine production/secretion by CD8 T cells (left). In contrast, disruption of EndoA3-mediated CIE or retromer-dependent retrograde transport in cancer cells impairs the efficient recruitment of immune synapse components to the contact area with CD8 T cells, resulting in a larger but less stable immune synapse. Although CD8 T cells attempt to compensate by expanding their contact areas with cancer cells, this is insufficient to support efficient cytokine production and secretion. However, the reduced stability of the immune synapse may facilitate more rapid detachment and re-engagement of CD8 T cells, thereby enhancing their lytic activity, potentially through increased serial killing (right). Data information: In (**A**), (**C**), and (**E**), images are representative of three independent experiments. In (**B**), (**D**), and (**F**), data are pooled from three independent experiments. Data are presented as median and quartiles. *p<0.05, **p<0.01, ***p<0.001 (**B**, Kolmogorov–Smirnov test; **D** and **F**, Mann–Whitney test).

The online version of this article includes the following source data and figure supplement(s) for figure 5:

**Figure supplement 1.** Inhibition of EndoA3-mediated endocytosis and retrograde transport affects ICAM1 recruitment to and structure of immune synapses.

**Figure supplement 1—source data 1.** Original files for western blot analysis displayed in *Figure 5—figure supplement 1B*.

**Figure supplement 1—source data 2.** PDF file containing original western blots for *Figure 5—figure supplement 1B*.

**Figure supplement 2.** Inhibition of EndoA3-mediated endocytosis and retrograde transport causes enlarged immune synapses.

*Figure 5 continued on next page*

*Figure 5 continued*

**Figure supplement 2—source data 1.** Original files for western blot analysis displayed in *Figure 5—figure supplement 2C and D*.

**Figure supplement 2—source data 2.** PDF files containing original western blots for *Figure 5—figure supplement 2C and D*.

*2015* on Fast Endophilin-Mediated Endocytosis (FEME) reported that the internalization of ICAM1 was not affected when all three EndoA isoforms were depleted. This discrepancy could be attributed to differences in the cell lines used. *Boucrot et al., 2015* used a human T cell line (Kit255), in which EndoA3 might be poorly expressed (*Giachino et al., 1997*; *Kjaerulff et al., 2011*), while we used human cancer cell lines – HeLa cells with endogenous EndoA3 expression and the LB33-MEL melanoma cell line engineered to stably express EndoA3. Moreover, it is now widely accepted that FEME is primarily mediated by the EndoA2 isoform (*Renard and Boucrot, 2021*).

Notably, all three confirmed EndoA3-dependent endocytic cargoes – ALCAM, L1CAM, and ICAM1 – are Ig-like cell adhesion molecules. This suggests the possibility of a common recognition pattern for EndoA3-mediated CIE. Unlike CME, where cargo recognition is typically mediated by adaptor protein binding to specific sorting signals in the cytosolic tails of the cargoes (*Traub and Bonifacino, 2013*), the mechanisms for cargo recognition in CIE are just beginning to emerge (*Lakshminarayan et al., 2014*; *Moreno-Layseca et al., 2021*). In FEME, EndoA2 is thought to directly recognize and sort G protein-coupled receptors (GPCRs) by binding to PRDs in GPCRs through their SH3 domains (*Boucrot et al., 2015*; *Tang et al., 1999*), or indirectly sort cargoes by interacting with other proteins, such as binding to CIN85 for EGFR and HGFR (*Soubeyran et al., 2002*; *Petrelli et al., 2002*), or srGAP1 for ROBO1 (*Genet et al., 2019*). However, how EndoA3-dependent endocytic cargoes are recognized and sorted remains unclear. Our lab has previously shown that the cytosolic tails of ALCAM and L1CAM physically interact with EndoA3, but not with EndoA2 (*Renard et al., 2020*; *Lemaigre et al., 2023*). However, specific interaction motifs have not yet been identified in these two cargoes, and it remains uncertain whether the interaction with EndoA3 is direct or indirect. If direct, it is still unknown which part of EndoA3 mediates this interaction. Nevertheless, the accumulating evidence that Ig-like cell adhesion molecules are preferentially EndoA3-dependent cargoes suggests the existence of a shared recognition pattern within this cargo family, warranting further investigations.

In addition, we found that two of the three confirmed EndoA3-dependent endocytic cargoes, ALCAM and ICAM1, are also retrograde transport cargoes dependent on the retromer complex. The fact that ICAM1 follows a retromer-dependent retrograde transport route aligns with previous proteomic analyses of retromer-dependent cargoes, where ICAM1 was identified among the hits (*Steinberg et al., 2013*). While the link between retrograde transport and cell polarity has been well-established (*Shafaq-Zadah et al., 2020*; *Shafaq-Zadah et al., 2016*; *Carpier et al., 2018*), the relationship between CIE and subsequent retrograde transport in polarized cellular contexts has largely been unexplored. Here, we propose that EndoA3-dependent cargoes, such as ICAM1 and ALCAM, use the retrograde transport route, thereby linking this specific CIE pathway to polarized cellular environments. Interestingly, our findings align with a previous study by Jo and colleagues which suggested that ICAM1 undergoes clathrin-independent endocytosis and is transported to the immune synapse from intracellular compartments in dendritic cells (*Jo et al., 2010*). In their cellular model, the authors also showed that this recycling of ICAM1 was LFA1-dependent, but this remains not known and to be further explored in our model system.

The retromer complex is one of the two complexes responsible for retrieving cargoes from endosomes for retrograde transport. The other complex, named Commander, consists of the CCC and retriever complexes. The CCC complex is made up of two coiled-coil domain-containing proteins CCDC22 and CCDC93, 10 COMMD (copper metabolism MURR1 [Mouse U2af1-rs1 region 1] domain) family members COMMD1-COMMD10, and DENND10 (differentially expressed in normal and neoplastic cells-containing protein 10, also called FAM45A) (*Healy et al., 2023*; *Mallam and Marcotte, 2017*; *Wan et al., 2015*; *Singla et al., 2019*). The retriever complex, structurally similar to the retromer complex, is composed of C16orf62 (VPS35L), DSCR3 (VPS26C), and VPS29 (*McNally et al., 2017*; *Chen et al., 2019*). Generally, the Commander complex is considered to be responsible for retromer-independent endosomal cargo sorting for retrograde transport, although many cargoes can utilize both complexes (*McNally et al., 2017*). Our study confirmed that both ALCAM and ICAM1 are retromer-dependent retrograde transport cargoes. However, while depletion of the retromer subunits VPS35 and VPS26A both impaired the retrograde transport of ALCAM and ICAM1

in HeLa cells, the effect of VPS26A depletion was consistently more pronounced than that caused by the loss of VPS35 (*Figure 1C and D*). Moreover, only VPS26A depletion impaired CD8 T cell activation, while VPS35 depletion had no significant effect (*Figure 3G*). Western blot analyses confirmed that VPS35 siRNA treatment led to a significant reduction in VPS26A levels, whereas VPS26A siRNA treatment resulted in a milder loss of VPS35 (*Figure 1C and D*, *Figure 1—figure supplement 1C and E*). This observation is in agreement with a previous report suggesting that VPS29 and VPS35 form a stable subcomplex in vivo, and that depletion of VPS35 or VPS29 leads to the degradation of other subunits, whereas VPS26A depletion has minimal effects on VPS29 and VPS35 levels (*Fuse et al., 2015*).

Based on these observations, we hypothesize that there may be a mutual compensatory relationship between the retromer and Commander complexes for some cargoes. Depletion of VPS35 in HeLa cells causes an almost complete loss of the retromer complex, potentially triggering compensation by the Commander complex. However, VPS26A depletion preserves the VPS29-VPS35 intermediate of the retromer complex (*Fuse et al., 2015*), which may inhibit compensation by the Commander complex through mechanisms that remain to be explored. This could explain why VPS35 depletion had a milder effect on the retrograde transport of ALCAM and ICAM1 (*Figure 1C and D*) and why it did not significantly affect CD8 T cell response (*Figure 3G*). Future investigations on the potential mutual compensation between the retromer and Commander complexes in the retrograde transport of Ig-like proteins will be necessary.

Moreover, a striking observation from our study is that disruption of EndoA3-mediated endocytosis in cancer cells leads to the formation of larger, dysfunctional contact zones with cytotoxic CD8 T cells, accompanied by reduced cytokine secretion, yet enhanced lytic activity. This counterintuitive result aligns with established evidence that full immune synapse maturation is not required for cytotoxicity. Rather, effective killing can occur through transient and serial interactions (*Purbhoo et al., 2004*). Indeed, when CD8 T cells fail to kill, they tend to form prolonged synapses and release higher levels of cytokines, whereas successful, rapid kills are associated with shorter, less stimulatory contacts (*Jenkins et al., 2015*). Single CD8 T cells, as well as CAR-T and NK cells, can kill multiple target cells sequentially through such transient contacts, so-called serial killing (*Davenport et al., 2015*; *Davenport et al., 2018*; *Cazaux et al., 2019*).

Notably, our results appear to contrast with the findings of *Petit et al., 2016*, who reported that larger immune synapses correlate with increased T cell adhesion to target cells, enhanced cytokine secretion, greater degranulation, and increased lytic activity. Taken together with our data, these observations indicate that although changes in immune synapse size always reflect alterations in T cell responses, synapse size alone does not allow one to infer synapse stability or transience, nor the magnitude of T cell activation. Therefore, an in-depth characterization of T cell responses – integrating multiple readouts such as cytokine production and secretion, surface expression of activation markers, proliferation, degranulation, and cytolytic activity – is essential.

Altogether, our findings suggest that by controlling the endocytic turnover and membrane delivery of adhesion molecules such as ICAM1 at the target-cell surface, EndoA3-mediated CIE tunes the signaling properties of the immune synapse. We propose a model (*Figure 5G*) where a lower EndoA3 level likely decreases ICAM1 redistribution, favoring larger but transient synapses that enhance cytolytic engagement with limited T cell cytokine secretion, which fits with the serial killing hypothesis. In contrast, active EndoA3-dependent endocytosis would promote faster adhesion-molecule turnover and formation of smaller, more stable synapses that enhance cytokine secretion by T cells but limit their serial killing capacity.

In conclusion, we identify ICAM1 as a novel EndoA3-dependent endocytic cargo, alongside ALCAM, and show that both rely on retromer-dependent retrograde transport. Our findings connect a defined clathrin-independent endocytic mechanism to the polarized recycling of immune synapse components, positioning EndoA3-mediated endocytosis as a potential regulatory axis that fine-tunes the balance between CD8 T cell cytokine production/secretion and lytic activity. In the longer term, pharmacological modulation of this pathway may represent an interesting avenue to modulate T cell-mediated anti-tumor immunity. However, translation into therapeutic applications remains distant and will require further investigation in future studies.

# Materials and methods

## Cell culture

Wild-type HeLa cell line was from ATCC. GalT-GFP-SNAP-expressing HeLa cell line was generated in a previous study (*Shi et al., 2012*). EndoA3-GFP-expressing HeLa cell line was previously generated by our lab (*Renard et al., 2020*). HLA-A*68012-expressing HeLa cell line was generated for the current study. Wild-type LB33-MEL cell line was derived from a melanoma patient in 1988 in de Duve Institute (*Lehmann et al., 1995*). EndoA3-GFP-expressing and GalT-GFP-SNAP-expressing LB33-MEL cell lines were generated for the current study. Anti-MUM-3 CD8 T cells were derived from the same melanoma patient in 1990 in de Duve Institute (*Lehmann et al., 1995*). Epstein–Barr virus-transformed B cells from cancer patient LG-2 (LG-2 EBV) were obtained as described (*Degiovanni et al., 1988*).

Wild-type HeLa cells were cultured at 37°C under 5% $CO_2$ in high glucose DMEM (Gibco, 41966-029) supplemented with 10% FBS (Dulis, 500105N1N), 100 U/mL penicillin, and 100 µg/mL streptomycin (Penicillin-Streptomycin, Gibco, 15140-122). EndoA3-GFP-expressing, GalT-GFP-SNAP-expressing, and HLA-A*68012-expressing HeLa cells were cultured at 37°C under 5% $CO_2$ in high glucose DMEM (Gibco, 41966-029) supplemented with 10% FBS, 100 U/mL penicillin, 100 µg/mL streptomycin, and 0.5 mg/mL G418 (Roche, 108321-42-2). WT LB33-MEL cells were cultured at 37°C under 5% $CO_2$ in IMDM (Gibco, 12440-053) supplemented with 10% FBS, 100 U/mL penicillin, 100 µg/mL streptomycin, 1.5 mM GlutaMAX I (Gibco, 35050-038), and 0.05 mM β-mercaptoethanol (Gibco, 21985-023). Empty plasmid stably transfected (Φ), EndoA3-GFP-expressing, and GalT-GFP-SNAP-expressing LB33-MEL cells were cultured in the same medium as WT LB33-MEL cells but supplemented with 0.7 mg/mL G418 (Roche, 108321-42-2). Anti-MUM-3 CD8 T cells were freshly thawed from cryovials and cultured at 37°C under 5% $CO_2$ in IMDM (Gibco, 12440-053) supplemented with 10% human serum (Ludwig Institute for Cancer Research (LICR), Brussels Branch), 100 U/mL penicillin, 100 µg/mL streptomycin, 1.5 mM GlutaMAX I, and extra 5 U/mL DNase I (STEMCELL Technologies, 07469) and 50 U/mL IL-2 (PROLEUKIN, CNK 1185–958) for at least 4 h before usage. For culture, T cells were maintained with the same medium, but without DNase I. LG-2 EBV cells were cultured at 37°C under 8% $CO_2$ in IMDM (Gibco, 21980-032) supplemented with 10% FBS (Gibco, A5256701), 100 U/mL penicillin, 100 µg/mL streptomycin, and 1.5 mM GlutaMAX I (Gibco, 35050-038).

All cell lines were regularly tested for mycoplasma and PIV-5 contamination using a luminescence detection kit (Lonza, LT07-318) and a PCR test, respectively. Only negative cells were used for experiments.

## DNA constructs and transfection

Plasmids used in this study are listed below:

| Protein | Backbone | Tag | Source |
|---|---|---|---|
| EndoA3 | pIRESneo2 | GFP | F. Tyckaert, LIBST, BE |
| / | pIRESneo3 | / | Clontech |
| ICAM1 | pLV-mScarlet | mScarlet | D. Van Buul, Amsterdam UMC, NL |
| ICAM1 | pEGFP-N1 | EGFP | Generated in the lab for this study |
| GalT-GFP-SNAP | pSNAP-tag | GFP | Johannes, Institut Curie, FR |
| / | pEGFP-N1 | EGFP | Clontech |
| HLA-A*68012 | pcDNA3 | / | van der Bruggen, de Duve Institute, BE |

C-terminally EGFP-tagged ICAM1 expression plasmid was generated in the current study for live-cell imaging. ICAM1 sequence was first amplified by PCR from the aforementioned mScarlet-tagged ICAM1 expression plasmid, and then inserted into the pEGFP-N1 vector by Gibson Assembly (New England Biolabs). To do so, the vector was linearized with the restriction enzyme EcoR I, and corresponding overlapping sequences were added to both ends of ICAM1 inserts by PCR (forward primer: 5′-CGAGCTCAAGCTTCGATGGCTCCCAGCAGC-3′; reverse primer: 5′-TACCGTCGACTG CAGGGGAGGCGTGGCTTG-3′). The Gibson assembly product was then amplified in *Escherichia coli* DH10B strain and validated by sequencing.

For immunofluorescence and live-cell imaging experiments, plasmids were transfected with FuGENE HD (Promega), according to the manufacturer's instructions. Cells were used for experiments 16–24 h after transfection.

To generate the stable HLA-A*68012-expressing HeLa cell line, the pcDNA3-HLA-A*68012 plasmid was transfected into wild-type HeLa cells by nucleofection (Amaxa Nucleofector 2b, Lonza) according to the built-in protocol. Transfected cells were cultured for 2 days. A selection pressure was then imposed by supplementing the culture medium with 1 mg/mL G418 for 7 days. Cells were then harvested and maintained with 0.5 mg/mL G418 in the culture medium.

For the generation of different stable LB33-MEL cell lines, 300,000 wild-type LB33-MEL cells were seeded in 2 mL culture medium. 4 µg of corresponding plasmid (pIRESneo3 for LB33-MEL Φ generation, pIRESneo2-EndoA3-GFP-FKBP for EndoA3-GFP-expressing LB33-MEL generation, and pGalT-GFP-SNAP-tag for GalT-GFP-SNAP-expressing LB33-MEL generation) were added to 200 µL of Opti-MEM (Gibco), followed by addition of 24 µL of FuGENE HD (Promega) and gentle mixing. The mix was incubated for 15 min at room temperature and then added to the cells. After 2 days of culture, transfected cells were selected by supplementing the culture medium with 0.7 mg/mL G418 for 7–15 days. Cell sorting based on GFP fluorescence was then conducted with the FACSAria flow cytometer (BD Biosciences) to yield a 100% positive stable EndoA3-GFP-expressing LB33-MEL cell line. For LB33-MEL cells transfected with GalT-GFP-SNAP-expressing plasmid, the homogeneity of cells was checked on a FACSVerse at the end of selection, and all cells were GFP-positive. Stable LB33-MEL Φ, EndoA3-GFP-expressing LB33-MEL and GalT-GFP-SNAP-expressing LB33-MEL cell lines were then maintained with continuous presence of 0.7 mg/mL G418 in the culture medium.

## RNA interference

siRNAs were used to knock down target molecules in HeLa and LB33-MEL cell lines at a total final concentration of 40 nM. For HeLa cells, 150,000 cells were seeded per well of a 6-well plate for each siRNA condition, and siRNA transfection was conducted with 1 µL Lipofectamine RNAiMAX (Thermo Fisher). For LB33-MEL cells, 200,000 cells were seeded per well of a 6-well plate for each siRNA condition, and siRNA transfection was conducted with 2 µL Lipofectamine RNAiMAX (Thermo Fisher). All experiments were done 72 h after siRNA transfection. All siRNAs were purchased from QIAGEN, and their sequences are listed as follows:

| Name | Reference | Sequence |
| --- | --- | --- |
| Hs_SH3GL3_5 | SI04170376 | CCAGACGAGAATACAAGCCAA |
| Hs_SH3GL3_6 | SI04176529 | CCAGACGAAGAAGTCAGACAA |
| Hs_VPS26A_1 | SI03122273 | AAAGGTAAACCTAGCCTTTAA |
| Hs_VPS26A_2 | SI04197914 | TGCCACCTATCCTGATGTTAA |
| Hs_VPS35_7 | SI04287605 | CAGAATTGCCCCTTAAGACTTT |
| Hs_VPS35_8 | SI04316914 | TTGCTGCATCCAAACTTCTAA |
| Rab6 HP Custom siRNA | NA | AAGACAUCUUUGAUCACCAGA |
| AllStars Negative Control | SI03650318 | NA |

## T cell proliferation assays and quantification of activation markers

For data of *Figure 3—figure supplement 1G–J*, siRNA-treated LB33-MEL EndoA3+ cells were detached with PBS-EDTA 2 mM, resuspended to 300,000 cells/mL, and γ-irradiated (100 Gy), as were LG2-EBV feeder cells. Anti-MUM-3 CD8 T cells were thawed the day before, labeled with 1 µM Cell-Trace Violet (Invitrogen, C34571) for 20 min at 37°C, washed, and resuspended to 300,000 cells/mL in T cell medium supplemented with 50 U/mL IL-2. In ultra-low attachment round-bottom 96-well plates (Corning Costar, CLS7007), 30,000 CD8 T cells were co-seeded with 15,000 LB33-MEL cells and 90,000 feeder cells. Plates were briefly centrifuged and incubated for 4 days at 37°C with 8% $CO_2$. Cells were then placed on ice for antibody staining and flow cytometry analysis.

For data of *Figure 4—figure supplement 1B*, HLA-A*68012 expressing HeLa cells were detached with PBS-EDTA 2 mM and resuspended to 1,000,000 cells/mL in the culture medium supplemented

with or without MUM-3 peptide (EAFIQPITR) at 300 ng/mL. Then, HeLa cells and LG-2 EBV feeder cells were γ-irradiated for 45 min (100 Gy), washed, and maintained in T cell culture medium until the initiation of co-culture with T cells. Anti-MUM-3 CD8 T cells were thawed the day before and were harvested, washed, and resuspended in complete T cell culture medium with 0.5 µM CMFDA (Invitrogen, C2925) for 20 min at 37°C. T cells were then washed and resuspended to 300,000 cells/mL in T cell medium supplemented with 50 U/mL IL-2. In an ultra-low attachment round bottom 96-well plate (Corning Costar, CLS7007), 30,000 anti-MUM-3 CD8 T cells were co-seeded with 10,000 HeLa cells and 90,000 γ-irradiated LG-2 EBV feeder cells. The plate was quickly centrifuged and then incubated for 4 days at 37°C with 8% $CO_2$. Cells were then transferred on ice for antibody staining and flow cytometry analysis.

Flow cytometry data were acquired with a LSR Fortessa (BD Biosciences) and analyzed using FlowJo (v10.8.1) software.

## T cell IFNγ secretion assay

CD8 T cells were thawed and cultured as described above. siRNA-transfected LB33-MEL cells were harvested 72 h after the siRNA transfection and were resuspended to 300,000 cells/mL in T cell medium. CD8 T cells were then washed and resuspended to 300,000 cells/mL in fresh T cell medium supplemented with 50 U/mL IL-2. 100 µL of LB33-MEL cells and 100 µL of T cells were pooled together in a 96-well plate and were co-cultured for 20 h at 37°C under 5% $CO_2$. A technical triplicate was made for each siRNA condition. The supernatant was collected the next day for ELISA analysis.

ELISA plate was coated with 50 µL IFNγ monoclonal coating antibody solution (Invitrogen, AHC4432; 1:250 dilution in PBS) in each well and was incubated at 4°C overnight. On the next day, two standard IFNγ concentration ladders were prepared: 1:2 dilutions for seven successive times with T cell medium in 96-well plate (original concentration of 10,000 pg/mL; the last well contains only medium). The ELISA plate was washed with washing buffer (0.154 M NaCl solution with 0.1% Tween-20) twice and with water once to remove unbound antibody. 50 µL of supernatant and standard IFNγ ladders were then added to the ELISA plate, followed by the addition of 50 µL IFNγ monoclonal detecting antibody solution (biotin conjugated; Invitrogen, AHC4539; 1:1000 dilution in T cell medium) into each well. The plate was incubated on a shaker at 37°C for 2 h and then was washed again. 50 µL of Streptavidin-Peroxidase solution (Sigma, S5512; 1:1000 dilution in PBS containing 0.5% bovine serum albumin) was added to each well and the plate was again incubated at 37°C for 30 min. Then, the plate was washed and 100 µL of 1-Step Ultra TMB-ELISA Substrate Solution (Thermo Fisher, 34028) was added to each well. When the blue color of TMB stopped getting darker, 20 µL of sulfuric acid (1 M) was added to each well to stop the reaction (the blue color turned yellow). The absorbance of substrate solution at 450 nm was then measured with an ELISA Analyzer (Bio-Rad Microplate Reader), and the IFNγ concentration was determined.

## Quantification of intracellular cytokines in CD8 T cells

CD8 T cells were thawed and cultured as described above. LB33-MEL cells were harvested 72 h after siRNA transfection and resuspended to 1,000,000 cells/mL in T cell medium. T cells were then washed and resuspended to 1,000,000 cells/mL in fresh T cell medium supplemented with 50 U/mL IL-2. 100 µL LB33-MEL cells and 100 µL T cells were pooled together in a 96-well 'U' bottom plate and were co-cultured at 37°C under 8% $CO_2$ for 1 h. Then, Brefeldin A (BFA; Sigma, B7651-5MG) was added to each well at a final concentration of 5 µg/mL. Co-culture continued for another 3 h, after which cells were transferred to a 96-well 'V' bottom plate and washed with 150 µL/well of staining buffer (PBS with 1 mM EDTA and 1% human serum). The plate was quickly spun at 400×$g$ for 3 min and supernatant was removed. 50 µL of extracellular staining antibody solution for CD8 and viability marker prepared with staining buffer were added to each well and mixed. The plate was then maintained at 4°C for 20 min in the dark. Afterwards, cells were washed with 150 µL/well PBS and resuspended in 100 µL/well of 1% paraformaldehyde (PFA) for 10 min at room temperature. Cells were then washed with PBS again and resuspended in 100 µL PBS containing 1 mM EDTA, 1% human serum, and 0.1% saponin. After 10 min incubation at room temperature, cells were centrifuged, and supernatant was removed. 50 µL of intracellular staining antibody solution for cytokines prepared with the same buffer was added to each well, and the plate was incubated in the dark for 30 min at room temperature. Finally, cells were washed and resuspended in 250 µL 1% PFA for flow cytometry analysis.

Flow cytometry data were acquired with a FACSVerse (BD Biosciences) and were analyzed using FlowJo (v10.8.1) software.

## T cell degranulation assay

CD8 T cells were thawed and cultured as described above. LB33-MEL cells were harvested 72 h after siRNA transfection and resuspended to 1,000,000 cells/mL in T cell medium. T cells were then washed and resuspended to 2,000,000 cells/mL in fresh T cell medium supplemented with 50 U/mL IL-2. 100 μL LB33-MEL cells and 50 μL T cells were pooled together in a 96-well 'U' bottom plate with 10 μL anti-CD107a antibody. Cells were co-cultured at 37°C under 8% $CO_2$ for 10 min and were then transferred on ice for extracellular marker and viability dye staining and flow cytometry analysis.

## T cell cytotoxicity assay

Anti-MUM-3 CD8 T cells were thawed one day before the experiment. siRNA-transfected LB33-MEL EndoA3+ cells were harvested 72 h post-transfection and resuspended to $1 \times 10^6$ cells/10 μL FBS. Cells were labeled with 20 μL Chromium-51 (Perkin Elmer, NEZ030S002MC) for 1 h at 37°C. After two centrifugations ($400 \times g$ for 8 min), cells were resuspended in 10 mL medium and incubated for an additional hour. After washing, labeled cells were counted and adjusted to 10,000 cells/mL. CD8 T cells were washed and resuspended to 300,000 cells/mL. In a 96-well V-bottom plate, 150 μL of T cells were dispensed and a threefold serial dilution was performed. Then, 100 μL labeled LB33-MEL cells were added, and plates were incubated for 4 h at 37°C with 8% $CO_2$. Supernatants (100 μL) were mixed with 150 μL scintillation fluid (Perkin Elmer, 6013117) in a chromium-reading plate (Perkin Elmer, 1450-40). Radioactivity was measured on a Microbeta2 counter (Perkin Elmer). Technical triplicates were included for each siRNA condition.

## Cell surface staining of cancer cells for flow cytometry

72 h after siRNA transfection, cells were detached and harvested with PBS-EDTA 4 mM and were transferred to pre-cooled 96-well 'V'-bottom plates with 100,000 cells in each well. The plate was centrifuged at $400 \times g$ for 4 min at 4°C. Cells were washed with 150 μL/well of staining buffer (PBS with 1 mM EDTA and 1% human serum) and centrifuged at $400 \times g$ for 4 min at 4°C. Cells were resuspended in 50 μL extracellular staining antibody solution for ALCAM and ICAM1 and viability marker, prepared with staining buffer. The plate was then maintained at 4°C for 20 min in darkness. Finally, cells were washed and resuspended in 250 μL 1% PFA for flow cytometry analysis.

Flow cytometry data were acquired with a FACSVerse (BD Biosciences) and were analyzed using FlowJo (v10.8.1) software.

## Western blotting

Cell lysates were prepared with 200,000 cells and 50 μL of sample buffer (62.5 mM Tris-HCl pH 6.8, 2% sodium dodecyl sulfate [SDS], 5% glycerol, 2.5% β-mercaptoethanol, 0.005% blue bromophenol). Lysate samples were boiled at 95°C for 10 min before loading onto 4–15% Mini-PROTEAN TGX precast protein gels (Bio-Rad, 4561084), as well as prestained protein standard. For the SNAP-Tag-based retrograde transport assay (see below), HiMark Pre-stained Protein Standard (Invitrogen, LC5699) was used; for all the other western blots, Broad Range (10–250 kDa) Color Prestained Protein Standard (New England Biolabs, P7719S) was used. After electrophoresis, proteins were transferred onto a PVDF membrane (pre-activated with methanol for 1 min; Millipore, IPFL85R) using a wet transferring system. The membrane was then blocked in 5%-milk PBS-T (phosphate-buffered saline, 0.1% Tween-20) for 1 h at room temperature, then stirred overnight at 4°C with primary antibody in the same solution. The next day, the membrane was incubated with corresponding horseradish peroxidase (HRP) conjugated secondary antibodies for 1 h at room temperature. Chemiluminescent (ECL) substrate (SuperSignal West Femto Maximum Sensitivity Substrate, Thermo Fisher, 34096, or SuperSignal West Pico PLUS Chemiluminescent Substrate, Thermo Fisher, 34580) was used for image exposure with an Amersham ImageQuant 800 machine (Cytiva). Images were processed and quantified with ImageJ v2.14.0/1.54f (NIH) software.

## SNAP-Tag-based retrograde transport assay

First, anti-ALCAM (Bio-Rad, MCA1926) and anti-ICAM1 (Bio-Rad, MCA1615) antibodies were labeled with benzylguanine (BG) by incubating them overnight at 4°C with a threefold molar excess of BG-G-LA-NHS reagent (New England Biolabs, S9151S; prepared in anhydrous DMSO). The reaction was neutralized with 20 mM Tris (pH 7.4) for 10 min at 21°C. Free BG-GLA-NHS was eliminated by applying labeled antibodies on spin desalting columns with a 7 kDa cutoff (Thermo Fisher, 89849), according to manufacturer's instructions. Then, retrograde transport assays were conducted according to previously published protocols (*Shafaq-Zadah et al., 2021*).

## Colocalization and uptake experiments

LB33-MEL cells were cultured on coverslips in 4-well plates to reach sub-confluence on the day of the experiment. Cells were pre-incubated in fresh serum-containing medium for 30 min at 37°C just before the experiment, then were incubated with anti-ALCAM antibody (BioLegend, 343902)-containing pre-warmed medium (5 µg/mL) for 15 min at 37°C to allow for endocytosis. For colocalization experiments, cells were quickly washed at 37°C with pre-warmed PBS and were subsequently fixed with pre-warmed 4% PFA for 20 min at 37°C in order to preserve the integrity of cellular structures and protein distribution. For uptake experiments, endocytosis was stopped on ice, and unbound antibodies were removed by extensive washes with ice-cold PBS++ (PBS supplemented with 0.4 mM $MgCl_2$ and 0.9 mM $CaCl_2$). Residual cell surface-accessible anti-ALCAM antibodies were stripped by two 5 s acid washes on ice (200 mM acetic acid, pH 2.5, 300 mM NaCl, 5 mM NaCl, 1 mM $CaCl_2$, and 1 mM $MgCl_2$). After neutralization by extensive washes with ice-cold PBS++, cells were fixed with 4% PFA for 10 min on ice, followed by another 10 min at room temperature. For both experiments, after quenching with 50 mM $NH_4Cl$ for at least 5 min, cells were permeabilized with 0.02% saponin in PBS containing 0.2% bovine serum albumin for 30 min at room temperature. After incubation with secondary antibodies in the same permeabilization solution for 30 min, samples were mounted with Fluoromount G (Invitrogen). Images were taken with a LSM900 microscope (Carl Zeiss) equipped with an Airyscan detector and a Plan Apo 63× numerical aperture (NA) 1.4 oil immersion objective. For colocalization experiments, Airyscan mode was applied. Pixel size: 0.04 micron. For uptake experiments, confocal mode was used. Pixel size: 0.07 micron. Microscope software: Zen Blue (v3.3; Carl Zeiss).

## Conjugate formation for confocal fixed sample imaging

For conjugate formation between adherent HLA-A*68012-expressing HeLa cells and anti-MUM-3 CD8 T cells (*Figure 4—figure supplement 1C*), HeLa cells were cultured on coverslips and transiently transfected with ICAM1-mScarlet expression plasmid 16–24 h before the experiment. HeLa cells were first incubated in 300 ng/mL MUM-3 peptide-containing culture medium for 45 min at 37°C and were then washed extensively with T cell culture medium to remove residual unbound peptide. T cells were freshly thawed as described above and resuspended to 500,000 cells/mL with T cell culture medium supplemented with 50 U/mL IL-2. Then, 250,000T cells were added to HeLa cells and the pool was incubated for 20 min at 37°C. Cells were then fixed at room temperature for 20 min by adding PFA directly to the culture medium to reach a final concentration of 4%.

For conjugate formation between suspended HLA-A*68012-expressing HeLa cells and anti-MUM-3 CD8 T cells (*Figure 5A*, *Figure 5—figure supplement 1A*), HeLa cells were cultured in 6-well plates and transfected with negative control siRNAs or siRNAs targeting EndoA3 72 h in advance and with plasmid expressing ICAM1-mScarlet 16–24 h in advance. HeLa cells were detached with PBS-EDTA 4 mM, resuspended to 1,000,000 cells/mL in culture medium containing 300 ng/mL MUM-3 peptide and incubated for 45 min at room temperature. Then, HeLa cells were extensively washed with T cell culture medium and finally resuspended to 1,000,000 cells/mL with T cell culture medium supplemented with 50 U/mL IL-2. T cells were freshly thawed as described above and resuspended to 1,000,000 cells/mL with T cell culture medium supplemented with 50 U/mL IL-2. Subsequently, 250,000 HeLa cells and 250,000T cells were pooled together and further co-cultured on coverslips for 20 min at 37°C. Of note, coverslips were pre-coated with poly-L-lysine (Sigma-Aldrich, P4832) according to manufacturer's instructions to stabilize conjugates. Cells were then fixed at room temperature for 20 min by adding PFA directly to the culture medium to reach a final concentration of 4%.

For conjugate formation between suspended LB33-MEL EndoA3+ cells and anti-MUM-3 CD8 T cells (*Figure 5C and E*, *Figure 5—figure supplement 2A and B*), LB33-MEL cells were transfected with negative control siRNAs or siRNAs targeting EndoA3 or VPS26A 72 h in advance. LB33-MEL cells were detached with 0.05% Trypsin-EDTA (Gibco, 25300-054) and resuspended to 1,000,000 cells/mL with T cell culture medium supplemented with 50 U/mL IL-2. T cells were freshly thawed as described above and resuspended to 1,000,000 cells/mL with T cell culture medium supplemented with 50 U/mL IL-2. Then, 250,000 LB33-MEL cells and 250,000T cells were pooled together and were co-cultured on poly-L-lysine pre-coated coverslips for 20 min at 37°C. Cells were then fixed at room temperature for 20 min by adding PFA directly to the culture medium to reach a final concentration of 4%.

After fixation with PFA and quenching with 50 mM NH₄Cl for at least 5 min, cells were permeabilized with 0.02% saponin in PBS containing 0.2% bovine serum albumin for 30 min, both at room temperature. Then, cells were successively incubated with primary and secondary antibodies in the same PBS-saponin solution for 30 min and mounted with Fluoromount G (Invitrogen). Images were taken with LSM900 microscope (Carl Zeiss) equipped with an Airyscan detector and a Plan Apo 63× numerical aperture (NA) 1.4 oil immersion objective. For the ICAM1 recruitment experiment (*Figure 4—figure supplement 1C*) and immune synapse morphology experiment (*Figure 5C and E*, *Figure 5—figure supplement 2A and B*), z-stack was used. z-stack interval: 0.15 micron. For all images, Airyscan mode was applied. Pixel size: 0.04 micron. Microscope software: Zen Blue (v3.3; Carl Zeiss).

## Conjugate formation for high-speed spinning-disk confocal live-cell imaging

HLA-A*68012-expressing HeLa cells were seeded in 8-well culture chamber slides (Ibidi, 80806) and transfected with ICAM1-mScarlet expression plasmid 16–24 h in advance. HeLa cells were first incubated in MUM-3 peptide-containing (300 ng/mL) culture medium for 45 min at 37°C and then washed extensively with T cell culture medium to remove residual unbound peptide. Anti-MUM-3 CD8 T cells were freshly thawed as described above and stained with CellTracker Green CMFDA (Invitrogen, C2925; 1:100 dilution in T cell culture medium supplemented with 50 U/mL IL-2) for 45 min at 37°C, washed with culture medium, and resuspended to 300,000 cells/mL in T cell culture medium supplemented with 50 U/mL IL-2. 100,000T cells were added to each well containing HeLa cells and live-cell imaging was started immediately after. Live-cell imaging was performed at 5% CO₂ and 37°C on an inverted Axio Observer 7 microscope (Carl Zeiss) equipped with a Yokogawa CSU-W1 spinning-disk confocal scanner unit, a sCMOS Photometrics Prime 95B camera, an α-Plan Apo 100× numerical aperture (NA) 1.46 oil immersion objective and a stage-top incubator. The microscope was controlled using Metamorph software (v7.10.4, Molecular Devices). Focus was maintained with Definite Focus 3 (Carl Zeiss). Several cells were imaged each for 3 min, with a time interval between each frame of 1 s. Pixel size: 0.11 micron.

## TIRF live-cell imaging

EndoA3-GFP-expressing HeLa/LB33-MEL cells were cultured on a 35 mm imaging dish with a glass bottom and low walls (ibidi, μ-Dish 35 mm, low, 80137) and transfected with ICAM1-mScarlet expression plasmid 16–24 h in advance. Live-cell imaging was conducted under 5% CO₂ and 37°C with Axio Observer 7 microscope (Carl Zeiss) equipped with an iLas azimuthal TIRF module (Gattaca Systems), an α Plan-Apo 100× NA 1.46 oil immersion objective, an sCMOS Prime BSI Express camera (Photometrics) and a stage-top incubator (Tokai Hit). The microscope was controlled using Metamorph software (v7.10.4.407; Molecular Devices). Laser focus was maintained with definite focus 3. Time interval between each frame: 1 s. Pixel size: 0.065 micron.

## Cancer cell area, aspect ratio, and roundness analysis by confocal imaging

For quantification of area, aspect ratio, and roundness of LB33-MEL EndoA3+ cells following EndoA3 depletion, cells were seeded on coverslips and transfected with siRNAs for 72 h. Cells were then fixed with 4% PFA for 20 min and a quenching step with 50 mM NH₄Cl was performed for at least 5 min. Cells were subsequently incubated with DAPI (Roche, 10236276001) and Phalloidin-Alexa Fluor 633 (Invitrogen, A22284, 1:500 dilution in a PBS solution containing 0.02% saponin and 0.2% bovine serum albumin) for 1 h at room temperature. Finally, coverslips were mounted with Fluoromount G

(Invitrogen). Images were acquired on a Cell Discoverer 7 (Carl Zeiss) in confocal mode, using a Plan Apo 5× numerical aperture (NA) 0.35 objective and a 2× tube lens. The microscope was controlled using Zen Blue (v3.7.97.09, Carl Zeiss). Pixel size: 0.282 micron.

## Microscope image analyses and quantifications

Airyscan image processing was performed with Zen Blue (v3.3; Carl Zeiss). Images were quantified with (Fiji Is Just) ImageJ v2.14.0/1.54f (NIH) software as follows.

For CD166 uptake experiments, images were quantified using Icy v2.4.2.0 (Institut Pasteur) software. Briefly, regions of interest (ROIs) were drawn manually around each cell. Within each ROI, bright spots corresponding to endocytic structures were automatically detected over dark background using the 'Spot Detector' plugin. The plugin performs image denoising by computing wavelet adaptive threshold (WAT) on the union of all ROIs present in an image. This automatic thresholding was then manually adjusted, depending on the size of the spots. For each independent experiment, scale 2 was chosen and sensitivity was adjusted empirically. In addition, a size filter was added to discard spots smaller than 2 pixels. After processing, an Excel file containing the fluorescence intensity of each endocytic structure was generated. The sum intensity of endocytic structures in each cell was calculated, then data were first adjusted according to the surface level of CD166 detected by flow cytometry in each experimental condition and were finally normalized to the control condition (set as 100%) for statistical analysis.

For conjugate formation experiments with HeLa cells to detect ICAM1 recruitment in the vicinity of CD8 T cells, masks were automatically generated for each image according to phalloidin staining by using Cellpose v3.0.7 software (*Stringer and Pachitariu, 2024*). First, an RGB format 512×512-pixel image of the phalloidin channel was generated from Fiji (ImageJ v2.14.0/1.54f) for each image and was imported to Cellpose. Segmentation of each cell was automatically performed using the 'cyto3' model with diameter (pixels) set as 40 for all independent repeats and was then exported as a mask. The mask was then imported into Fiji and re-scaled to the real size of the original image. Wand tool was used to select ROI corresponding to single cells. In each ROI, the raw integrated density (RawIntDen) of ICAM1 signal was measured. Then, the ICAM1 recruitment ratio in the vicinity of the T cell was calculated as the ratio between the ICAM1 signal in the T cell ROI and the total ICAM1 signal in the ROIs of the T cell and the adjacent HeLa cell: 'RawIntDen of ICAM1 signal in T cell ROI/(RawIntDen of ICAM1 signal in T cell ROI +RawIntDen of ICAM1 signal in HeLa cell ROI)'. Data were normalized to control condition (set as 100%) for each independent repeat and were then pooled together for statistical analysis.

For conjugate formation experiments to measure the size of immune synapses, z-stack images (z-stack interval: 0.15 micron) for each conjugate were imported to Fiji. The contact region between a CD8 T cell and an LB33-MEL cell was delineated by the 'freehand line' tool manually, and then the length of the line was measured for each z-stack. The largest value was picked out from all z-stacks and was considered as the size of this immune synapse. All data from three independent repeats were then pooled together for statistical analysis.

Tracking of ICAM1-positive carriers in live-cell spinning-disk images of HeLa cells upon formation of immune synapses with CD8 T cells was performed using Fiji software (ImageJ v2.14.0/1.54f). Segmentation of moving carriers was first carried out by machine learning using the Trainable Weka Segmentation plugin (*Arganda-Carreras et al., 2017*). Tracking was then performed with TrackMate (*Tinevez et al., 2017*), using the Advanced Kalman Tracker. Resulting tracks were filtered based on their duration, displacement, and quality to refine the dataset and remove non-moving structures. Track counts within defined cellular regions were calculated and normalized to the surface area of each region to obtain track densities. Track displacements and directions (for polar plot) were computed separately in Python using a custom script (see *Source code 1*).

For quantification of area, aspect ratio, and roundness of LB33-MEL EndoA3+ cells upon EndoA3 depletion, Arivis software (v4.1.1, Carl Zeiss) was used. Cells were segmented using Cellpose (*Stringer et al., 2021*) Cyto2 model based on phalloidin and DAPI staining.

## Statistical analyses

Statistical analyses were performed with GraphPad Prism v10.1.0. D'Agostino–Pearson omnibus normality test was used to check the normality of datasets. If data passed the normality test, parametric

tests were used, and data were plotted on graphs as mean ± SEM as error bars. If data did not pass the normality test, nonparametric tests were used, and data were plotted on graphs as median and quartiles. Details on the parametric and nonparametric tests used for each analysis are indicated in figure legends. Significance of comparison is represented on the graphs by asterisks.

## Antibodies and reagents

| Product | Reference | RRID | Application | Dilution |
|---|---|---|---|---|
| PE anti-human CD166 antibody | BioLegend, 343904 | AB_2289302 | FACS | 1:1000 |
| APC anti-human CD54 antibody | BD Pharmingen, 561899 | AB_10896118 | FACS | 1:60 |
| APC anti-HLA-A2, A28 antibody | Miltenyi Biotec, 130-099-582 | AB_2652041 | FACS | 1:300 |
| Viability Dye eFluor 780 | Invitrogen, 65-0865-14 | / | FACS | 1:1000 |
| PerCP-Cy5.5 anti-human CD8 antibody | BD Pharmingen, 560662 | AB_1727513 | FACS | 1:100 |
| APC anti-human IL-2 antibody | BioLegend, 500310 | AB_315097 | FACS | 1:25 |
| BV421 anti-human TNF-α antibody | BioLegend, 502932 | AB_10960738 | FACS | 1:100 |
| PE anti-human IFNγ antibody | BD Pharmingen, 554701 | AB_395518 | FACS | 1:200 |
| Biotin anti-human HLA-A2 Antibody | BioLegend, 343322 | AB_2572191 | FACS | 1:400 |
| PE-Cy7 anti-human CD107a | BioLegend, 328618 | AB_11147955 | FACS | 1:20 |
| BV785 anti-human PD-1 antibody | BioLegend, 329930 | AB_2563443 | FACS | 1:40 |
| PE anti-human CD137 antibody | BioLegend, 309804 | AB_314783 | FACS | 1:200 |
| PE-Cy7 anti-human TIM-3 | BioLegend, 345014 | AB_2561720 | FACS | 1:50 |
| CMFDA | Invitrogen, C2925 | / | FACS | 0.5 µM |
| Anti-IFNγ coating antibody | Invitrogen, AHC4432 | AB_2536280 | ELISA | 1:250 |
| Biotin anti-IFNγ detecting antibody | Invitrogen, AHC4539 | AB_2536281 | ELISA | 1:1000 |
| Streptavidin−Peroxidase | Sigma-Aldrich, S5512 | / | ELISA | 1:1000 |
| 1-Step Ultra TMB-ELISA Substrate | Thermo Fisher, 34028 | / | ELISA | 100 µL |
| Anti-SH3GL3/Endophilin-A3 antibody | Sigma-Aldrich, HPA039381 | AB_10794635 | WB | 1:1000 |
| Anti-α-tubulin antibody | Sigma-Aldrich, T5168 | AB_477579 | WB | 1:10000 |
| Anti-VPS26A antibody | Abcam, ab23892 | AB_2215043 | WB | 1:1000 |
| Anti-VPS35 antibody | Abcam, ab10099 | AB_296841 | WB | 1:500 |
| HRP conjugated anti-mouse antibody | Agilent Technologies, P044701-2 | / | WB | 1:5000 |
| HRP conjugated anti-rabbit antibody | Agilent Technologies, P044801-2 | / | WB | 1:5000 |
| HRP conjugated anti-goat antibody | Santa Cruz Biotechnology, sc-2354 | AB_628490 | WB | 1:1000 |
| Anti-SNAP-tag antibody | New England Biolabs, P9310S | AB_10631145 | WB | 1:1000 |
| Anti-Rab6A antibody | Abcam, ab95954 | AB_10679758 | WB | 1:1000 |
| Anti-CD166 antibody | Bio-Rad, MCA1926 | AB_323338 | SNAP-Tag | 10 µg/mL |
| Anti-CD54 antibody | Bio-Rad, MCA1615 | AB_321783 | SNAP-Tag | 10 µg/mL |
| SNAP-Cell Block | New England Biolabs, S9106S | / | SNAP-Tag | 1:200 |
| Anti-CD166 antibody | BioLegend, 343902 | AB_2223892 | IF | 1:100 |
| Anti-CD45 antibody | Santa Cruz Biotechnology, sc-1178 | AB_627074 | IF | 1:100 |
| SNAP-Cell TMR-Star | New England Biolabs, S9105S | / | IF | 1:200 |
| Anti-GM130 antibody | BioLegend, 937002 | AB_2888895 | IF | 1:200 |

*Continued on next page*

*Continued*

| Product | Reference | RRID | Application | Dilution |
|---|---|---|---|---|
| Alexa Fluor 488 Phalloidin | Invitrogen, A12379 | / | IF | 1:200 |
| Alexa Fluor 568 Phalloidin | Invitrogen, A12380 | / | IF | 1:200 |
| Alexa Fluor 488 anti-Rabbit IgG | Invitrogen, A11008 | AB_143165 | IF | 1:200 |
| Alexa Fluor 568 anti-Mouse IgG | Invitrogen, A11031 | AB_144696 | IF | 1:200 |
| Alexa Fluor 633 anti-Mouse IgG | Invitrogen, A21050 | AB_2535718 | IF | 1:200 |
| CellMask Plasma Membrane Stain | Invitrogen, C10045 | / | IF | 1:1000 |
| DAPI | Sigma-Aldrich, D9542 | / | IF | 1:10000 |
| Poly-L-Lysine | Sigma-Aldrich, P4832 | / | IF | NA |

## Acknowledgements

We thank JD van Buul (Amsterdam UMC, Netherlands) for kindly providing the aforementioned plasmid. We thank E Rigaux, F Tyckaert, C Wildmann, C Duhamel, S Meurant, A Fattaccioli, S Burteau, and C Demazy for their support and technical help in experiments. We thank E Macdonald for proofreading the manuscript. We greatly thank the Morph-Im platform of UNamur and the flow cytometry & cell sorting platform (CYTF) from the de Duve Institute of UCLouvain for giving access to advanced microscopy and flow cytometry technologies, respectively. SX is supported by a PhD fellowship from FSR (Fond Spécial de Recherche) of UNamur and UCLouvain. AB is a FRIA (Fonds pour la Formation à la Recherche dans l'Industrie et l'Agriculture) PhD Research Fellow from the Fonds de la Recherche Scientifique (FNRS, Belgium). TH is a postdoctoral research fellow of the FNRS (Belgium). LT is supported by a Marie Skłodowska-Curie postdoctoral fellowship (grant agreement n° 101151524) under the European Union's Horizon 2020 program, and by an Honorary Postdoctoral Research Fellowship of the FNRS (Belgium) and an additional operating grant from the King Baudouin Foundation (Belgium). LJ is supported by Mizutani Foundation for Glycosciences (reference n° 200014), Agence National de la Recherche (ANR-20-CE15-0009-01, ANR-22-CE11-0030-03, France), Fondation pour la Recherche Médicale (EQU202103012926, France) and an ERC Proof of Concept (project 101062030). PVDB and TH were supported by de Duve Institute (Belgium) and Université Catholique de Louvain (Belgium). PM is supported by a PDR (T.0163.21) and CDR (J.0127.23) from the Fonds de la Recherche Scientifique (FNRS, Belgium). H-FR is supported by a Start-Up Grant Collen-Francqui from Francqui Foundation (Belgium), an Incentive Grant for Scientific Research (MIS-F.4540.21) and a Research Credit (CDR-J.0176.24) from the Fonds de la Recherche Scientifique (FNRS, Belgium), and a research grant from NARILIS institute (UNamur).

## Additional information

### Funding

| Funder | Grant reference number | Author |
|---|---|---|
| UNamur and UCLouvain, Fonds Spéciaux de Recherche | PhD Fellowship | Shiqiang Xu |
| Fonds pour la Formation à la Recherche dans l'Industrie et dans l'Agriculture | PhD Fellowship | Alix Buridant |
| Fonds De La Recherche Scientifique - FNRS | Postdoc Fellowship | Thibault Hirsch Louise Thines |

| Funder | Grant reference number | Author |
|---|---|---|
| H2020 Marie Skłodowska-Curie Actions | 101151524 | Louise Thines |
| King Baudouin Foundation | | Louise Thines |
| Mizutani Foundation for Glycoscience | 200014 | Ludger Johannes |
| Agence Nationale de la Recherche | ANR-20-CE15-0009-01 | Ludger Johannes |
| Agence Nationale de la Recherche | ANR-22-CE11-0030-03 | Ludger Johannes |
| Fondation pour la Recherche Médicale | EQU202103012926 | Ludger Johannes |
| European Research Council | 101062030 | Ludger Johannes |
| Fonds De La Recherche Scientifique - FNRS | T.0163.21 | Pierre Morsomme |
| Fonds De La Recherche Scientifique - FNRS | J.0127.23 | Pierre Morsomme |
| Fondation Francqui - Stichting | Start-Up Grant | Henri-François Renard |
| Fonds De La Recherche Scientifique - FNRS | MIS-F.4540.21 | Henri-François Renard |
| Fonds De La Recherche Scientifique - FNRS | CDR-J.0176.24 | Henri-François Renard |
| de Duve Institute (Belgium) | | Pierre Van der Bruggen Thibault Hirsch |
| Université Catholique de Louvain (Belgium) | | Pierre Van der Bruggen Thibault Hirsch |
| UNamur, NARILIS (Belgium) | | Henri-François Renard |

The funders had no role in study design, data collection and interpretation, or the decision to submit the work for publication.

## Author contributions

Shiqiang Xu, Conceptualization, Data curation, Formal analysis, Investigation, Visualization, Methodology, Writing – original draft, Writing – review and editing; Alix Buridant, Céline Duhamel, Formal analysis, Investigation, Visualization; Thibault Hirsch, Conceptualization, Formal analysis, Supervision, Investigation, Methodology, Writing – review and editing; Benjamin Ledoux, Resources, Software, Formal analysis, Validation, Investigation, Visualization, Methodology, Writing – review and editing; Massiullah Shafaq-Zadah, Estelle Dransart, Methodology, Writing – review and editing; Louise Thines, Formal analysis, Investigation, Visualization, Methodology, Writing – review and editing; Ludger Johannes, Conceptualization, Funding acquisition, Writing – review and editing; Pierre Van der Bruggen, Pierre Morsomme, Conceptualization, Supervision, Funding acquisition, Project administration, Writing – review and editing; Henri-François Renard, Conceptualization, Supervision, Funding acquisition, Investigation, Methodology, Writing – original draft, Project administration, Writing – review and editing

## Author ORCIDs

Shiqiang Xu ⓘD https://orcid.org/0000-0002-0696-499X
Alix Buridant ⓘD https://orcid.org/0009-0000-4549-0013
Thibault Hirsch ⓘD https://orcid.org/0000-0002-4448-1948
Benjamin Ledoux ⓘD https://orcid.org/0009-0004-0336-6478
Massiullah Shafaq-Zadah ⓘD https://orcid.org/0000-0002-7582-8131
Estelle Dransart ⓘD https://orcid.org/0000-0003-2547-6825

Louise Thines https://orcid.org/0009-0003-4208-4580
Ludger Johannes https://orcid.org/0000-0002-2168-0004
Pierre Van der Bruggen https://orcid.org/0000-0002-3910-4101
Pierre Morsomme https://orcid.org/0000-0001-7780-7230
Henri-François Renard https://orcid.org/0000-0002-2406-2519

Reviewer #1 (Public review): https://doi.org/10.7554/eLife.105821.3.sa1
Reviewer #2 (Public review): https://doi.org/10.7554/eLife.105821.3.sa2
Reviewer #3 (Public review): https://doi.org/10.7554/eLife.105821.3.sa3
Author response https://doi.org/10.7554/eLife.105821.3.sa4

## Additional files

### Supplementary files

MDAR checklist

Source code 1. Custom Python script for quantifying track displacements and directions of ICAM1-positive carriers from live-cell spinning-disk microscopy of HeLa cells.

### Data availability

Source data for western blots are provided with this paper as separate source data files. All numerical data used to generate the corresponding graphs and perform statistical analyses in each figure are available in the public repository Dryad: https://doi.org/10.5061/dryad.bcc2fqzts.

The following dataset was generated:

| Author(s) | Year | Dataset title | Dataset URL | Database and Identifier |
|---|---|---|---|---|
| Xu S, Buridant A, Hirsch T, Duhamel C, Ledoux B, Shafaq-Zadah M, Dransart E, Thines L, Johannes L, van der Bruggen P, Morsomme P, Renard H-F | 2026 | Clathrin-independent endocytosis and retrograde transport in cancer cells tune immune synapse organization and CD8 T cell response | https://doi.org/10.5061/dryad.bcc2fqzts | Dryad digital repository, 10.5061/dryad.bcc2fqzts |

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
