## [Editor Report · eLife Assessment]

This study now provides **solid** evidence for a role of EndoA3-mediated trafficking of ICAM-1 to the immune synapse with T cells. The study will be **valuable** to those studying cell-cell communication in the immune system, and opens additional questions regarding the mechanisms involved and how other adhesion ligands are regulated.

---

## [Referee Report · Reviewer #1 (Public review)]

Summary:

This study by Xu et al. investigates how clathrin-independent endocytosis in cancer cells influences T cell activation. Using a combination of biochemical approaches and imaging, the authors identify ICAM1, the ligand for the T cell integrin LFA-1, as a novel cargo of EndoA3-mediated endocytosis.

The authors then explore the functional consequences of EndoA3 depletion in cancer cells on T cell function using cytokine measurements, surface marker analyses, cytotoxicity assays and imaging. Loss of EndoA3 results in reduced T cell cytokine production, while expression of activation and exhaustion markers such as TIM-3, PD-1, and CD137 remains largely unchanged. EndoA3 knockout is associated with reduced ICAM1 surface levels and increased ALCAM levels in cancer cells. Imaging experiments further reveal directional transport of ICAM1 toward the immunological synapse, seemingly slightly reduced ICAM1 levels at the synapse upon EndoA3 depletion and an enlarged contact area between T cells and cancer cells.

Based on these observations, the authors propose a model in which EndoA3-mediated endocytosis and retrograde trafficking of ICAM1 (and ALCAM) supplies the immunological synapse with ligands for adhesion molecules. In the absence of EndoA3, T cells are suggested to compensate for suboptimal ICAM1 availability by enlarging the synaptic contact area, altering synapse architecture, leading to reduced cytokine secretion but modestly enhanced cytotoxicity.

Overall, the study provides convincing evidence for a modulatory role of EndoA3-mediated endocytosis in regulating T cell-cancer cell interactions. However, the choice of cellular model systems, the limited number of biological replicates and insufficiently supported mechanistic interpretations weaken the manuscript and weaken the strength of its conclusions.

Strengths:

The authors employ a rigorous and innovative experimental strategy that convincingly identifies ICAM1 as a novel cargo of EndoA3-mediated endocytosis with convincing visualization of directional ICAM1 transport toward the immunological synapse. In addition, the study provides a comprehensive characterization of how EndoA3 depletion in cancer cells affects T cell cytokine production, activation, proliferation and cytotoxic function, representing a valuable contribution to our understanding of how membrane trafficking pathways in target cells can modulate immune responses.

Comments on revised version:

Thank you very much for submitting your revised manuscript. I appreciated your efforts to answer all of the reviewers questions. While in my opinion the manuscript truly improved I think there are still lingering questions, in particular regarding the following points:

(1) Limited biological replication:

The LB33-MEL system remains problematic, as also noted by other reviewers. While it clearly represents an improvement over highly derived model systems such as Jurkat or Raji cells, it nevertheless effectively restricts the study to a single biological replicate. In this context, it may be more appropriate to compare the chosen approach to more state-of-the-art systems, such as expression of HLA-A*02:01, peptide loading (e.g. NY-ESO), and introduction of the matching TCR into donor-derived primary T cells. Such an approach would allow the use of multiple T cell donors and would substantially strengthen the generalizability of the conclusions.

(2) Expression levels of ICAM1:

Based on available database information (e.g. UniProt) and published literature (PMID: 9371813), ICAM1 appears to be expressed at relatively low levels in both HeLa and LB33-MEL cells. While the effects on T cells are initially discussed in terms of broader changes in EndoA3-mediated recycling of multiple surface proteins, including ICAM1 and ALCAM (and potentially others), the focus of the manuscript increasingly shifts toward ICAM1 as the primary driver of the observed phenotypes. Given the comparatively low endogenous expression of ICAM1 in the chosen model systems, it is unclear whether this emphasis is fully justified. In addition, if ICAM1 polarization toward the immunological synapse was assessed using ICAM1 overexpression, whereas other phenotypes (such as enlarged contact area) were analyzed under endogenous expression conditions, this further complicates the interpretation. As a first step toward clarifying these issues, it would be helpful to include representative flow cytometry histograms showing surface expression levels of ICAM1 and ALCAM, rather than only normalized quantifications.

(3) Cell-cell contact dynamics:

The manuscript suggests that altered contact dynamics may underlie the observed increase in cytotoxicity upon EndoA3 depletion. However, these claims are not directly tested. Such effects could be addressed with relatively straightforward experiments, for example by directly measuring T cell-cancer contact duration in co-culture assays.

---

## [Referee Report · Reviewer #2 (Public review)]

The manuscript by Xu et al. studies the relevance of endophilin A3-dependent endocytosis and retrograde transport of immune synapse components and in the activation of cytotoxic CD8 T cells. First, the authors show that ICAM1 and ALCAM, known component of immune synapses, are endocytosed via endoA3-dependent endocytosis and retrogradely transported to the Golgi. The authors then show that blocking internalization or retrograde trafficking reduces the activation of CD8 T cells. Moreover, this diminished CD8 T cells activation resulted the formation of an enlarged immune synapse with reduced ICAM1 recruitment.

Comments on revisions:

The authors have addressed all my comments adequately.

---

## [Referee Report · Reviewer #3 (Public review)]

Shiqiang Xu and colleagues have examined the importance of ICAM-1 and ALCAM internalization and retrograde transport in cancer cells on formation of a polarized immunological synapse with cytotoxic CD8+ T cells. They find that internalization is mediated by Endophilin A3 (EndoA3) while retrograde transport to the Golgi apparatus is mediated by the retromer complex. Perturbing these trafficking pathways reduces cytokine release, but increases cytolytic killing. The paper is building on previous findings from corresponding author Henri-François Renard showing that ALCAM is an EndoA3 dependent cargo in clathrin-independent endocytosis.

The work is interesting as it describes a novel mechanism by which cancer cells might influence CD8+ T cell activation and immunological synapse formation, and the authors have used a variety of cell biology and immunology methods to study this. The authors have also made substantial efforts to address the reviewers comments to the first version of the paper. However, there are still some points which could be further improved to underpin their conclusions:

The movies and the related micrographs of EndoA3-mediated ICAM-1 endocytosis could be more convincing. Is the invagination of large membrane patches visible by volumetric imaging (e.g. confocal z-stacks) or brightfield microscopy?

There is still a lack of quantitative evidence for polarized transport of ICAM-1 positive vesicles towards the immunological synapse. Only one example is shown and the authors state that the data is from a single movie representative of two independent experiments. If there are multiple cells per experiment, the number of cells should be stated and more examples should be included.

---

## [Author Response]

The following is the authors’ response to the original reviews.

**Public Reviews:**

**Reviewer #1 (Public review):**
Summary:This study by Xu et al. focuses on the impact of clathrin-independent endocytosis in cancer cells on T cell activation. In particular, by using a combination of biochemical approaches and imaging, the authors identify ICAM1, the ligand for T cell-expressed integrin LFA-1, as a novel cargo for EndoA3-mediated endocytosis. Subsequently, the authors aim to identify functional implications for T cell activation, using a combination of cytokine assays and imaging experiments.They find that the absence of EndoA3 leads to a reduction in T cell-produced cytokine levels. Additionally, they observe slightly reduced levels of ICAM1 at the immunological synapse and an enlarged contact area between T cells and cancer cells. Taken together, the authors propose a mechanism where EndoA3-mediated endocytosis of ICAM1, followed by retrograde transport, supplies the immunological synapse with ICAM1. In the absence of EndoA3, T cells attempt to compensate for suboptimal ICAM1 levels at the synapse by enlarging their contact area, which proves insufficient and leads to lower levels of T cell activation.Strengths:The authors utilize a rigorous and innovative experimental approach that convincingly identifies ICAM1 as a novel cargo for Endo3A-mediated endocytosis.Weaknesses:The characterization of the effects of Endo3A absence on T cell activation appears incomplete. Key aspects, such as surface marker upregulation, T cell proliferation, integrin signalling and most importantly, the killing of cancer cells, are not comprehensively investigated.

We agree with the reviewer that the effects of EndoA3 depletion on T cell activation were not characterized enough. In new data presented in Fig.S4G-J, we explored additional activation markers and proliferation parameters. We didn’t observe any difference for the surface markers PD-1, CD137 and Tim-3 between LB33-MEL EndoA3+ cells treated with control and EndoA3 siRNAs. Regarding proliferation (Fig. S4J), although the proliferation index seems slightly lower upon EndoA3 depletion, we didn’t observe any significant difference either. Degranulation has also been monitored (Fig. S4K), but we didn’t observe any significant differences. In the new Fig. 3F however, we performed chromium release assays to assess the killing of cancer cells. Very interestingly, we observed an ~15% higher lysis of LB33-MEL EndoA3+ cells after EndoA3 depletion, when compared to the control condition at a ratio of 3:1 T cells:target cells (where the maximal effect is observed). These data are further discussed in the discussion section (new §6-9).

As Endo- and exocytosis are intricately linked with the biophysical properties of the cellular membrane (e.g. membrane tension), which can significantly impact T-cell activation and cytotoxicity, the authors should address this possibility and ideally address it experimentally to some degree.

Evaluating changes in the biophysical properties of cancer cell plasma membrane upon EndoA3 depletion is not trivial. An indirect way to address this question is by observing the area and shape of cells after siRNA treatment. In the new data added in the new Fig. S4B-D, we compared the area, aspect ratio and roundness of LB33-MEL EndoA3+ cells treated with negative control or EndoA3 siRNAs. While we observed a slight cell area reduction upon EndoA3 depletion, no significant changes were observed regarding the aspect ratio and the roundness. Hence, we think that the biophysical properties of cancer cells are not drastically modified by EndoA3 depletion.

Crucially, key literature relevant to this research, addressing the role of ICAM1 endocytosis in antigen-presenting cells, has not been taken into consideration.

We thank the reviewer for this important point. We have now considered and cited the relevant literature (Discussion, Page no.9).

**Reviewer #2 (Public review):**
Summary:The manuscript by Xu et al. studies the relevance of endophilin A3-dependent endocytosis and retrograde transport of immune synapse components and in the activation of cytotoxic CD8 T cells. First, the authors show that ICAM1 and ALCAM, known components of immune synapses, are endocytosed via endoA3-dependent endocytosis and retrogradely transported to the Golgi. The authors then show that blocking internalization or retrograde trafficking reduces the activation of CD8 T cells. Moreover, this diminished CD8 T cell activation resulted in the formation of an enlarged immune synapse with reduced ICAM1 recruitment.Strengths:The authors show a novel EndoA3-dependent endocytic cargo and provide strong evidence linking EndoA3 endocytosis to the retrograde transport of ALCAM and ICAM1.Weaknesses:The role of EndoA3 in the process of T cell activation is shown in a cell that requires exogenous expression of this gene. Moreover, the authors claim that their findings are important for polarized redistribution of cargoes, but failed to show convincingly that the cargoes they are studying are polarized in their experimental system. The statistics of the manuscript also require some refinement.

We fully acknowledge that the requirement for exogenous expression of EndoA3 in our immunological model represents a limitation of our study. Unfortunately, it remains challenging to identify cancer cell lines for which autologous CD8 T cells are available and that endogenously express all molecular players investigated (in particular EndoA3). At this stage, we do not have access to any other cancer cell line/autologous CD8⁺ T cell pairs that are sufficiently well characterized. In future studies, it would be valuable to investigate tumor types with high endogenous EndoA3 expression (such as glioblastomas, gliomas, and head and neck cancers) for which autologous CD8 T cells could be obtained, but this remains technically challenging.

To address the reviewer’s second point regarding polarized redistribution of cargoes, we have added new data in the new Figure 4 and Movies S8-9. Using high-speed spinningdisk live-cell confocal microscopy, we captured the movement of ICAM1-positive tubulovesicular carriers in cancer cells at the moment of contact with CD8 T cells. Capturing such events is technically challenging, as T cell–cancer cell contacts form randomly and transiently. Successful imaging requires that the cancer cell be well spread and express ICAM1–GFP at an optimal level (as it is transiently expressed as a GFP-tagged construct), while acquisition must occur precisely at the moment when the T cell initiates contact. Despite these technical constraints, we successfully imaged early stages of immune synapse formation, enabling visualization of ICAM1 vesicular transport.

The data reveal a flux of ICAM1-positive carriers emerging from the perinuclear region (corresponding to the Golgi area) and moving toward the contact site with the CD8 T cell, with fusion events of vesicles occurring at the developing immune synapse. AI-based segmentation and tracking analyses showed that ICAM1-positive carrier trajectories were predominantly oriented toward the forming immune synapse, whereas carriers moving toward other cellular regions were markedly less frequent. These results provide direct evidence for polarized ICAM1 transport via vesicular trafficking toward the immune synapse.

**Reviewer #3 (Public review):**
Summary:Shiqiang Xu and colleagues have examined the importance of ICAM-1 and ALCAM internalization and retrograde transport in cancer cells on the formation of a polarized immunological synapse with cytotoxic CD8+ T cells. They find that internalization is mediated by Endophilin A3 (EndoA3) while retrograde transport to the Golgi apparatus is mediated by the retromer complex. The paper is building on previous findings from corresponding author Henri-François Renard showing that ALCAM is an EndoA3dependent cargo in clathrin-independent endocytosis.Strengths:The work is interesting as it describes a novel mechanism by which cancer cells might influence CD8+ T cell activation and immunological synapse formation, and the authors have used a variety of cell biology and immunology methods to study this. However, there are some aspects of the paper that should be addressed more thoroughly to substantiate the conclusions made by the authors.Weaknesses:In Figure 2A-B, the authors show micrographs from live TIRF movies of HeLa and LB33MEL cells stably expressing EndoA3-GFP and transiently expressing ICAM-1-mScarlet. The ICAM-1 signal appears diffuse across the plasma membrane while the EndoA3 signal is partially punctate and partially lining the edge of membrane patches. Previous studies of EndoA3-mediated endocytosis have indicated that this can be observed as transient cargo-enriched puncta on the cell surface. In the present study, there is only one example of such an ICAM-1 and EndoA3 positive punctate event. Other examples of overlapping signals between ICAM-1 and EndoA3 are shown, but these either show retracting ICAM1 positive membrane protrusions or large membrane patches encircled by EndoA3. While these might represent different modes of EndoA3-mediated ICAM-1 internalization, any conclusion on this would require further investigation.

We agree with the reviewer that the pattern of cargoes during endocytosis (puncta vs large patches) as observed by live-cell TIRF microscopy may be confusing. Actually, a punctate pattern has been observed quasi systematically when we monitored the uptake of endogenous cargoes via antibody uptake assays (whatever the imaging approach: TIRF, spinning-disk, classical confocal or lattice light-sheet microscopy). For example:

- ALCAM: Fig.1e-h, Supplementary Figure 5 and Supplementary Movies 1-3 and 6 in Renard et al. 2020, https://doi.org/10.1038/s41467-020-15303-y; Fig.1D and Movie 2 in Tyckaert et al. 2022, https://doi.org/10.1242/jcs.259623.

- L1CAM: Fig.2 and 3D, Movies S1-4 in Lemaigre et al. 2023, https://doi.org/10.1111/tra.12883.

In rare examples, bigger clusters of antibodies were observed, where EndoA3 was observed to surround them, delineate them in a “lasso-like” pattern, and the clusters were progressively taken up:

- ALCAM: Supplementary Movie 4 in Renard et al. 2020, https://doi.org/10.1038/s41467-020-15303-y.

However, bigger patches of cargoes were more often observed when uptake was observed using transient expression of GFP-/mCherry-tagged versions of cargoes. In these cases, EndoA3 was predominantly observed to delineate cargo patches as a “lasso-like” pattern, progressively triming those patches leading to endocytosis. For example:

- L1CAM: Fig.3E, Movie S5-7 in Lemaigre et al. 2023, https://doi.org/10.1111/tra.12883.

- We also observed this pattern with CD166-GFP (unpublished).

The fact that we observed rather patches than punctate patterns upon transient expression of fluorescently-tagged constructs of cargoes is likely due to the elevated expression level of the cargoes.

Therefore, the patchy pattern observed for ICAM1 and ALCAM, transiently expressed in fusion with fluorescent proteins, and surrounded by EndoA3 in Fig.2A-B and old Movies S1-3, is not surprising. Of note, upon anti-ALCAM antibody uptake, we observed a more punctate pattern (Fig.2C), as previously described. Unfortunately, the lower quality of commercial anti-ICAM1 antibody did not allow us to proceed to uptake assays as for ALCAM.

Regarding Fig.S2 and old Movies S4-5, we agree with the reviewer that these data may be misleading, as they represent phenomena happening at protrusions and contact zones between two adjacent cells. We have now replaced these images with other examples where we avoid contact zones (Fig.S2 and new Movies S5-7).

These different patterns (patches vs dots) are still unexplained at the current stage, and may indeed represent different modes of endocytosis. We think these various patterns may depend on the abundance/expression level of cargoes and their degree of clustering. This will be investigated in future studies. Still, whatever the pattern, these data demonstrate and confirm the association between EndoA3 and cargoes (such as ICAM1 or ALCAM), even in the absence of antibodies.

Moreover, in Figure 2C-E, uptake of the previously established EndoA3 endocytic cargo ALCAM is analyzed by quantifying total internal fluorescence in LB33-MEL cells of antibody labelled ALCAM following both overexpression and siRNA-mediated knockdown of EndoA3, showing increased and decreased uptake respectively. Why has not the same quantification been done for the proposed novel EndoA3 endocytic cargo ICAM-1? Furthermore, if endocytosis of ICAM-1 and ALCAM is diminished following EndoA3 knockdown, the expression level on the cell surface would presumably increase accordingly. This has been shown for ALCAM previously and should also be quantified for ICAM-1.

As correctly pointed by the reviewer, anti-ICAM1 antibody uptake assays would have been great. We have tried to do them many times. Unfortunately, all commercial antibodies we tested did not yield satisfying results in uptake experiments. Either the labeling was too week/non-specific, or the antibody was not effectively stripped from the cell surface by acid washes, i.e. the acid-wash conditions required for efficient stripping were too harsh for the cells to tolerate. We have tried other approaches using the same commercial antibody which do not require acid washes (loss of surface assays by FACS, or uptake assays using surface protein biotinylation) or based on insertion of an Alfa-tag in the extracellular part of ICAM1 by CRISPR-Cas9 and detection of ICAM1 with an antiAlfa-tag nanobody (unpublished approach; collaboration with the lab of Prof. Leonardo Almeida-Souza, University of Helsinki, who developed the approach), but without success. However, we were more successful with the SNAP-tag-based approach to follow retrograde transport, for which the commercial anti-ICAM1 antibody worked properly. In Fig. 1F, we could show that retrograde transport of ICAM1 (and thus most likely its endocytosis step) was significantly decreased upon EndoA3 depletion in HeLa cells, indirectly demonstrating that ICAM1 is effectively an EndoA3-dependent cargo.

Regarding the fact that surface level of ICAM1 should increase upon perturbation of EndoA3-mediated endocytosis, we agree with the reviewer that this could be an expected result. However, this is not necessarily systematic, as the surface level of a protein cargo is always the result of a balance between its endocytosis, recycling to plasma membrane, and lysosomal degradation. We also have to take into account the neosynthesized protein flux. One must also consider that multiple endocytic mechanisms exist in parallel, and that the perturbation of one mechanism (EndoA3-mediated CIE, here) may be partially compensated by others, as cargoes can often be taken up via multiple endocytic doors. Hence, an increased abundance at the cell surface is not always guaranteed upon endocytosis perturbation. Anyway, we measured the cell surface level of both ICAM1 and ALCAM in LB33-MEL EndoA3+ cells treated with negative control or EndoA3 siRNAs (Fig. S4E-F). Only minor differences were observed.

In Figure 4A the authors show micrographs from a live-cell Airyscan movie (Movie S6) of a CD8+ T cell incubated with HeLa cells stably expressing HLA-A*68012 and transiently expressing ICAM1-EGFP. From the movie, it seems that some ICAM-1 positive vesicles in one of the HeLa cells are moving towards the T cell. However, it does not appear like the T cell has formed a stable immunological synapse but rather perhaps a motile kinapse. Furthermore, to conclude that the ICAM-1 positive vesicles are transported toward the T cell in a polarized manner, vesicles from multiple cells should be tracked and their overall directionality should be analyzed. It would also strengthen the paper if the authors could show additional evidence for polarization of the cancer cells in response to T-cell interaction.

A similar point was raised by reviewer #2. We have revised this section accordingly. In the new Fig. 4 and Movies S8-9, we replaced the live-cell Airyscan confocal data with highspeed spinning-disk confocal imaging data, enabling a more accurate analysis of cargo polarized redistribution and at a higher time resolution.

Using this approach, we captured the movement of ICAM1-positive tubulo-vesicular carriers in cancer cells at the moment of contact with CD8 T cells. Capturing such events is technically challenging, as T cell–cancer cell contacts form randomly and transiently. Successful imaging requires that the cancer cell be well spread and express ICAM1–GFP at an optimal level (as it is transiently expressed as a GFP-tagged construct), while acquisition must occur precisely at the moment when the T cell initiates contact. Despite these technical constraints, we successfully imaged early stages of immune synapse formation, enabling visualization of ICAM1 vesicular transport.

The data reveal a flux of ICAM1-positive carriers emerging from the perinuclear region (corresponding to the Golgi area) and moving toward the contact site with the CD8 T cell, with fusion events of carriers occurring at the developing immune synapse.

AI-based segmentation and tracking analyses showed that ICAM1-positive carrier trajectories were predominantly oriented toward the forming immune synapse, whereas carriers moving toward other cellular regions were markedly less frequent. These results provide direct evidence for polarized ICAM1 transport via vesicular trafficking toward the immune synapse.

Finally, in Figures 4D-G, the authors show that the contact area between CD8+ T cells and LB33-MEL cells is increased in response to siRNA-mediated knockdown of EndoA3 and VPS26A. While this could be caused by reduced polarized delivery of ICAM-1 and ALCAM to the interface between the cells, it could also be caused by other factors such as increased cell surface expression of these proteins due to diminished endocytosis, and/or morphological changes in the cancer cells resulting from disrupted membrane traffic. More experimental evidence is needed to support the working model in Figure 4H.

Regarding the cell surface expression of both ICAM1 and ALCAM, as already explained above, only minor differences were observed (Fig. S4E-F). Regarding morphological changes of cancer cells upon EndoA3 depletion (Fig. S4B-D), we compared the area, aspect ratio and roundness of LB33-MEL EndoA3+ cells treated with negative control or EndoA3 siRNAs. While we observed a slight cell area reduction upon EndoA3 depletion, no significant changes were observed regarding the aspect ratio and the roundness. Cancer cell morphology is thus not drastically modified by EndoA3 depletion. All these new data are now discussed in the manuscript.

**Recommendations for the authors:**

**Reviewing Editor Comments:**
The reviewers discussed the paper and all agreed it was incomplete in supporting the conclusions. Additional data needed to support the conclusions were:(1) Better characterisation of Endo3A-expressing and knock-down cells such as morphology, ICAM-1, and ALCAM surface levels to name two parameters.

As discussed above, we have now added new data addressing these points:

- Morphology: Fig. S4B-D

- ICAM1 and ALCAM surface levels: Fig. S4E-F These new data are discussed in the main text.

(2) Better characterisation of the ICAM-1 polarisation process. Does this require interaction with LFA-1 can ICAM-1 be delivered to the synapse without this?

As discussed above, we have now added new data better addressing the characterization of ICAM1 polarized trafficking to the immune synapse, that can be found in the new Fig. 4 (high-speed spinning-disk confocal imaging of ICAM1 trafficking upon conjugate formation between CD8 T cell and cancer cell). The text has been modified accordingly. The dependency on LFA-1 has not been addressed directly, but we may suppose it is indeed important as (i) it has already been addressed in other cellular systems by previous studies (Jo et al. 2010), and (ii) we observed a denser flux of ICAM1-positive carriers in the cancer cell toward regions involved in immune synapses with CD8 T cells, than other regions. As we didn’t address this question more directly in our study, we briefly mentioned this point in the Discussion section.

(3) Better characterisation of T cell response- activation markers, cytotoxicity assays.

As discussed above, we have now added new data addressing these points:

- Cell surface activation markers: Fig. S4G-I

- Proliferation: Fig. S4J

- Degranulation: Fig. S4K

- Cytotoxic activity: Fig. 3F

These new data are discussed in the main text.

(4) Citing relevant literature.

The relevant literature (in particular the paper by Jo et al. 2010) is now cited and discussed.

(5) Number of donors evaluated - is it true there was only one blood donor? For human studies better to have key results on >4 donors.

Our immunological working model indeed originates from a single patient (Baurain et al., 2000), from whom both a cancer cell line (LB33-MEL) and autologous CD8 T cells were derived. These CD8 T cells specifically recognize an HLA molecule presenting a defined antigenic peptide (MUM-3) on the surface of the cancer cells. This provides us with a unique and fully natural experimental system that allows us to faithfully reconstitute cytotoxic T lymphocyte (CTL)-mediated killing of cancer cells in vitro.

Using CD8 T cells from other donors would not be meaningful in this context, as they would not recognize the LB33-MEL cells. Conversely, testing the same CD8 T cells on other cancer cell lines requires engineering these lines to express the appropriate HLA molecule and to be exogenously pulsed with the correct antigenic peptide – which is precisely what we did with the HeLa cell line.

Therefore, increasing the number of donors would require obtaining both cancer cell lines and CD8 T cells from each donor, ideally with evidence that the donor’s T cells recognize their own tumor cells. This is technically challenging and not trivial, although it would indeed be highly valuable to diversify immunological models in future studies.

Importantly, the high specificity of our autologous co-culture system, where cancer cells interact with their naturally matched CD8 T cells, offers clear advantages over commonly used in vitro models such as Jurkat (T) and Raji (B) cell lines, which rely on artificial stimulation with a superantigen to enforce immunological synapse formation and T cell activation.

(6) How does the binding of antibodies to ICAM-1 and ALCAM impact their trafficking?

As IgG antibodies are bivalent and can bind two target antigens, they may induce clustering, which could in turn affect endocytosis. To address this concern, we performed an uptake assay based on surface protein biotinylation using a cleavable biotin reagent (with a reducible linker). Briefly, after allowing endocytosis for different time intervals, cell surface–exposed biotins were removed by treatment with the cellimpermeable reducing agent MESNA, while internalized (endocytosed) biotinylated proteins remained protected. These internalized proteins were then recovered by affinity purification on streptavidin resin and analyzed by Western blot to detect the protein of interest.

Importantly, this uptake assay can be performed in the absence or presence of an anticargo antibody, allowing assessment of its potential influence on endocytosis. Author response image 1 shows the results for ALCAM uptake in HeLa cells, with and without anti-ALCAM antibody:

**Author response image 1. sa4fig1:** Antibody binding to an extracellular epitope of ALCAM increases its endocytosis. HeLa cellsurface proteins were biotinylated on ice using EZ-Link Sulfo-NHS-SS-Biotin (Pierce) and then incubated at 37 °C for the indicated times to allow endocytosis. Internalization was assessed in the absence or presence of an anti-ALCAM antibody (Ab) added to the extracellular medium. Endocytosis was stopped by returning the cells to ice, and surface-exposed biotin was removed by treatment with the cell-impermeable reducing agent MESNA. Internalized, MESNA-resistant biotinylated proteins were affinity-purified on streptavidin resin and analyzed by Western blot to detect ALCAM. The “unstripped” condition shows the total amount of ALCAM at the cell surface at the beginning of the experiment (signal at ~95 kDa). Quantification of the time course (normalized to the no-antibody condition) shows increased ALCAM endocytosis in the presence of antibody at 15 and 30 min. Blot is representative of two independent experiments; quantifications include data from both experiments.

We observed that the anti-ALCAM antibody slightly enhanced ALCAM uptake. A similar experiment was attempted for ICAM1, but we were unable to detect the protein by Western blot using the available commercial antibody.

Although this outcome was expected, it highlights a potential caveat in using antibodies to monitor endocytosis. Alternative tools such as nanobodies, while monovalent and theoretically less perturbing, are not yet available for many cargo proteins and may still influence cargo conformation or dynamics. Therefore, antibodies remain the current gold standard in endocytosis studies. Nevertheless, data obtained with antibodies should always be validated by complementary approaches that do not rely on antibody binding, as we have done in this study (e.g. live-cell imaging of fluorescently tagged proteins).

The work is of interest and we look forward to your response/revision.
**Recommendations for the authors:**

**Reviewer #1 (Recommendations for the authors):**
Thank you for submitting your manuscript which I had the pleasure to review. While I enjoyed your work, I feel that it would strongly benefit by addressing the following points:(1) In-depth characterization of T cell responses upon Endo3A depletion: The characterization should be expanded to include surface marker upregulation, T cell proliferation, and, most importantly, tumor cell cytotoxicity. I was wondering if the incomplete characterization of T-cell responses is due to limited supplies of antigenspecific T-cells? My understanding is that these cells have been derived from a single patient. This also raises concerns in terms of reproducibility as all data are practically from a single biological replicate. My suggestion would be to use an additional system of specific cell-cell contacts to complement the current findings. For instance, HeLa cells could be transfected to express CD19 or EpCAM, for both of which bispecific T cell engagers (Invivogen) exist that would allow specific contact formation, thereby allowing the study of the effect of Endo3A depletion across T cells from different donors and through a more complete set of assays.

We refer the reviewer to our responses above, where these points have been addressed in detail. We sincerely thank the reviewer for the excellent suggestion of transfecting HeLa cells with CD19 or EpCAM and using bispecific T-cell engagers. However, after careful consideration, we concluded that this approach falls outside the scope of the present study, which was specifically designed to investigate the most natural system, cancer cells and their autologous CD8 T cells. We nevertheless appreciate this insightful suggestion and will certainly consider it for future studies.

(2) Alterations in membrane tension as an alternative explanation: Endo- and exocytosis have been found to influence the biophysical properties of cells, such as membrane tension (e.g., Djakbaravo et al., 2021, PMID: 33788963), which in turn influences their susceptibility to cytotoxic T cells with lower tension corresponding to reduced cytotoxicity (e.g., Basu & Whitlock, 2016, PMID: 26924577). Thus, interference with endocytic pathways could arguably lead to changes in membrane tension that could contribute to the observed effects. These possible effects should be discussed and addressed experimentally to a degree. While measuring membrane tension directly requires specialized expertise (e.g., tether pulling experiments) and is not within the scope of this study, membrane tension affects cell spreading and actin organization. Thus, I would suggest conducting a thorough comparative phenotypical and morphological characterization of the Endo3A+ and Endo3A- cancer cells to estimate the possible effect of changes in membrane tension (if any) on the results.

We refer the reviewer to our responses above, where these points have been addressed in detail. New data have been added and the text of our manuscript has been modified accordingly.

(3) Citation and consideration of earlier work: Jo & Kwon et al., 2010 (PMID: 20681010) have previously shown that ICAM1 undergoes clathrin-independent recycling and repolarization to the immunological synapse in APCs. Furthermore, they provided evidence that actin-based transport, but not lateral diffusion, together with recycling is crucial for the repolarization of ICAM1 to the immunological synapse. This important earlier work has to be cited. Actin-based transport on the cell surface has not been considered in the current manuscript. In light of these earlier findings, it is unclear in Figure 4A if ICAM1 is delivered to the T cell from within- or from the surface of the cancer cell. I would suggest changing the imaging modalities in this experiment to be able to differentiate cell surface from internal ICAM1, e.g., by detaching the cancer cells from the surface as has been done in Fig. 4B, E, and F.

We refer the reviewer to our responses above, where these points have been addressed in detail. New data have been added and the text of our manuscript has been modified accordingly.

**Reviewer #2 (Recommendations for the authors):**
Major comments:(1) The authors should be more careful with their claims about the importance of their results for cell polarity as their evidence for this is scarce (i.e. The live-cell imaging in Figure 4A is not quantified and the ICAM1 polarization effect shown in figure 4B-C is, albeit significant, small and not very convincing).

We refer the reviewer to our responses above, where these points have been addressed in detail. New data have been added and the text of our manuscript has been modified accordingly.

(2) The absence (or very low expression) of EndoA3 on the LB33-MEL cell suggests that EndoA3-mediated recycling of immune synaptic components is not required for T-cell activation. The fact that EndoA3 exogenous expression in LB33-MEL cells leads to increased cytokine production in T cells is, however, interesting.

We fully agree with the reviewer’s observation. Although EndoA3 is not expressed in some cellular contexts, its cargoes may still be present. It is therefore reasonable to assume that alternative endocytic mechanisms can compensate for its absence. It is now widely accepted that many cargoes can be internalized through multiple endocytic routes, and that the relative contribution of each pathway depends strongly on the cellular and physiological context.

For example, we have shown that ALCAM and L1CAM, although primarily internalized via clathrin-independent pathways, present a minor fraction (< 25%) undergoing clathrinmediated endocytosis (Renard et al., 2020; Lemaigre et al., 2023). Moreover, we observed that inhibition of macropinocytosis enhances EndoA3-mediated endocytosis of ALCAM, indicating a crosstalk between specific EndoA3-mediated clathrin-independent endocytosis (CIE) and non-specific macropinocytosis (Tyckaert et al., 2022).

Thus, even in the absence of EndoA3, its cargoes are likely internalized through alternative endocytic routes. Nonetheless, our data clearly demonstrate that EndoA3 expression markedly enhances the endocytosis and intracellular trafficking of its cargoes, ultimately leading to modified CD8 T cell responses.

(3) For the statistics in bar graphs (graphs 1C, D, E &F; 3E, 3F, S1C-I, and S3C), one cannot have all values for controls simply normalized to 1. This procedure hides the variance for the controls between each replicate and makes any statistics meaningless.

We thank the reviewer for this important remark. Regarding Figures 1C–F, S1C–I, and S3C, which correspond to quantifications from Western blots, it is standard practice to normalize the quantification to a control condition set to 1 (or 100%). Absolute signal intensities cannot be directly compared across different blots due to the variability inherent to this semi-quantitative technique. For this reason, we chose to keep the data presented in normalized form. However, we agree that this type of data require the careful choice of a convenient statistical analysis approach. Here, we choose one-sample T tests, allowing to test the hypothesis that the various siRNA conditions are different from 100% (the normalized value of the siCtrl condition). We adapted the statistical analysis accordingly in the different figures mentioned.

Regarding old Figures 3E–F (now Fig. 3E and 3G), which correspond to IFNγ secretion assays, we agree that representing IFNγ secretion as a fold change relative to a control condition may obscure inter-experimental variability. However, this format was intentionally chosen to facilitate data interpretation, as IFNγ secretion was quantified by ELISA and also displayed inter-experimental variability. For completeness, we now provide below the corresponding graphs showing absolute IFNγ concentrations, which retain the information on inter-experimental variability (Author response image 2). As you can see, the overall conclusions remain unchanged.

**Author response image 2. sa4fig2:** IFNg secretion data corresponding to Fig. 3E and 3G, expressed in absolute values (pg/mL)

Minor comments:(1) What happens to surface and total levels of ICAM1 and ALCAM in the retromer or EndoA3 knockdown/overexpression conditions? This information would put the effects described into context.

We refer the reviewer to our responses above, where these points have been addressed in detail. New data have been added and the text of our manuscript has been modified accordingly.

(2) The authors should clearly indicate that BFA means bafilomycin A in the figure legend or methods.

BFA corresponds to Brefeldin A. We have now clarified this information in legends and methods.

(3) In the sentence: "These data demonstrate that retromer-mediated retrograde transport is critical for trafficking ALCAM and ICAM1 to the Golgi and that this process requires the full secretory capacity of the TGN." What do the authors mean by full secretory capacity?

We have modified the sentence: “Together, these data demonstrate that retromermediated retrograde transport is critical for trafficking ALCAM and ICAM1 to the Golgi and that this process requires efficient secretion from the TGN (as evidenced by the involvement of Rab6).”

(4) The method used for retrograde transport seems to be a variation of the original protocol (reference 43). The manuscript would benefit from a thorough explanation of this assay, rather than citing the original protocol.

We did not modify the original SNAP-tag–based protocol used to monitor retrograde transport. A comprehensive methodological paper has been published (ref. 44), and we have followed it strictly. Additionally, we briefly summarized the rationale of the approach in Figure 1A and in the first paragraph of the Results section.